# Hybrid Latent Reasoning via Reinforcement Learning

**Zhenrui Yue[1], Bowen Jin[1], Huimin Zeng[1], Honglei Zhuang[2], Zhen Qin[2], Jinsung Yoon[2],**
**Lanyu Shang[3], Jiawei Han[1], Dong Wang[1]**
[1]University of Illinois Urbana-Champaign, [2]Google, [3]LMU
{zhenrui3,bowenj4,huiminz3,lshang3,hanj,dwang24}@illinois.edu,
{hlz,zhenqin,jinsungyoon}@google.com, lanyu.shang@lmu.edu

## Abstract

Recent advances in large language models (LLMs) have introduced latent reasoning as a promising alternative to autoregressive reasoning. By performing internal computation with hidden states from previous steps, latent reasoning benefit from more informative features rather than sampling a discrete chain-of-thought (CoT) path. Yet latent reasoning approaches are often incompatible with LLMs, as their continuous paradigm conflicts with the discrete nature of autoregressive generation. Moreover, these methods rely on CoT traces for training and thus fail to exploit the inherent reasoning patterns of LLMs. In this work, we explore latent reasoning by leveraging the intrinsic capabilities of LLMs via reinforcement learning (RL). To this end, we introduce hybrid reasoning policy optimization (HRPO), an RL-based hybrid latent reasoning approach that (1) integrates prior hidden states into sampled tokens with a learnable gating mechanism, and (2) initializes training with predominantly token embeddings while progressively incorporating more hidden features. This design maintains LLMs' generative capabilities and incentivizes hybrid reasoning using both discrete and continuous representations. In addition, the hybrid HRPO introduces stochasticity into latent reasoning via token sampling, thereby enabling RL-based optimization without requiring CoT trajectories. Extensive evaluations across diverse benchmarks show that HRPO outperforms prior methods in both knowledge- and reasoning-intensive tasks. Furthermore, HRPO-trained LLMs remain interpretable and exhibit intriguing behaviors like cross-lingual patterns and shorter completion lengths, highlighting the potential of our RL-based approach and offer insights for future work in latent reasoning.

## 1 Introduction

Latent reasoning has emerged as a compelling alternative to traditional autoregressive reasoning methods in large language models (LLMs) [8, 35, 39]. In contrast to the conventional chain-of-thought (CoT) [43, 17, 10], which relies on the discrete decoding and sampling process, latent reasoning enables LLMs to reason internally with continuous hidden representations from the previous steps. For instance, Coconut [11] achieves latent reasoning by utilizing the model's last hidden state as 'continuous thought', feeding it back as input embeddings to the next reasoning step, thereby matching the performance of CoT on reasoning-intensive tasks. To show the difference between the autoregressive generation and latent reasoning, we compare both approaches in Figure 1.

Nevertheless, existing methods in latent reasoning utilize extensive CoT traces for training. That is, CoT trajectories are required to learn informative latent representations. An example is CODI [35], which adopts self-distillation to train on discrete CoT tokens and transfers learnt features into continuous thoughts. Although recurrent latent reasoning removes the need for CoT data, it relies on training a multi-block LLM from scratch to reason internally [8]. Moreover, these methods employ tailored training paradigms for latent representation learning, incurring high training costs

39th Conference on Neural Information Processing Systems (NeurIPS 2025).

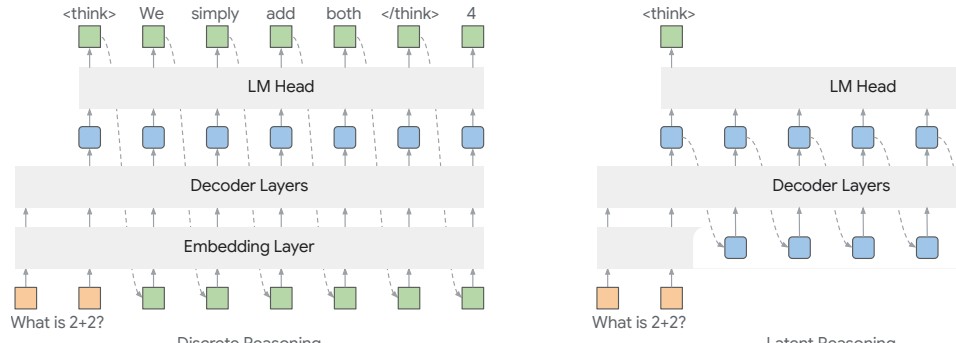

Figure 1: Comparison between discrete reasoning (left) and latent reasoning (right). Unlike the autoregressive sampling process in discrete reasoning, latent reasoning incorporates hidden representations from previous steps to enhance reasoning performance (between `<think>` and `</think>`).

and overlooking the inherent reasoning capabilities of LLMs [11, 8, 34]. For example, Coconut [11] requires multi-stage training on CoT steps, which not only increases training compute but also delays the model's acquisition of complete reasoning chains [35]. Furthermore, we find that latent reasoning is often incompatible with LLMs due to the discrepancy between output hidden states and input embeddings (as we show Section 4.3). That is, feeding hidden states into the next decoding step degrades generation quality (e.g., repetition, incoherence), causing difficulties in adapting LLMs for latent reasoning. Therefore, an ideal latent reasoning method should capitalize on pretrained LLMs' generalizability by seamlessly integrating continuous representations, preserving LLMs' interpretability while mitigating CoT-dependent extensive training for broader applicability.

To this end, we introduce hybrid reasoning policy optimization (HRPO), a novel hybrid latent reasoning optimization framework based on reinforcement learning (RL). HRPO unifies policy learning with latent reasoning, thereby utilizing the LLMs' intrinsic reasoning patterns without relying on CoT trajectories. To preserve the generative capabilities while encouraging the model to reason in the continuous space, HRPO introduces a gating mechanism to gradually incorporate hidden state representations from previous steps into sampled token embeddings. The gating mechanism is initially configured in a way that the inputs come predominantly from the sampled tokens. As training progresses, the gate learns to incorporate richer, more informative features from previous hidden states for improved internal reasoning. Since the sampling operation introduces stochasticity, HRPO rollouts can be performed like standard RL methods, with hybrid outputs (tokens and latent representations) stored in the rollout buffer for policy updates. For optimization, HRPO leverages a simple outcome-based reward and employs the hybrid rollout buffer to calculate log probabilities, enabling policy gradient updates that adaptively integrate both token-level and latent representations. By bridging discrete and continuous reasoning, HRPO provides a scalable and training-efficient solution that unlocks latent reasoning in existing LLMs. As a result, HRPO enhances the adaptability of latent reasoning and leads to superior performance on both knowledge- and reasoning-intensive tasks. We highlight our contributions in the following[1]:

- We introduce HRPO, the first reinforcement learning-based approach for hybrid reasoning, empowering LLMs to autonomously develop latent reasoning capabilities.

- We design a gating mechanism to preserve LLMs' generative abilities, which starts by prioritizing sampled token embeddings and, through RL-driven updates, progressively incorporates the continuous representations.

- By leveraging the LLMs' inherent reasoning patterns through HRPO, we mitigate the need for chain-of-thought annotations and expensive multi-stage training, offering an efficient and scalable alternative to existing latent reasoning methods.

- To show the efficacy of the proposed hybrid latent reasoning, we evaluate on multiple knowledge and reasoning benchmarks and show that it outperforms existing models and latent reasoning baselines, demonstrating consistent performance gains across diverse scenarios.

---

[1]Our implementation is available at https://github.com/Yueeeeeeee/HRPO.

In addition, we provide insights into RL-based training of latent reasoning models and present intriguing reasoning patterns emerging from HRPO.

## 2 Related Work

### 2.1 Latent Reasoning

Early research in latent reasoning focuses on analyzing the latent space computation within transformer models [2, 47]. For example, Biran et al. [2] study multi-hop reasoning and show that 'back-patch' features from later layers can improve performance on challenging queries. Alternatively, latent representations can be used to construct informative features as in-context demonstrations to enhance few-shot performance at test-time [45, 52]. In particular, Xu et al. [45] exploit latent skills to select in-context examples for reasoning-intensive tasks. Different from this line of work, hidden reasoning is also proposed to improve generative capabilities by incorporating latent variables into language modeling [8, 20]. For instance, Geiping et al. [8] propose a depth-recurrence language model that injects latent variables and iteratively processes them to derive the final states used for decoding. Similarly, special tokens (e.g. <pause>) are inserted to allocate extra test-time compute for internal reasoning, leading to improvements across diverse scenarios [9, 29]. Pfau et al. [29] argue that filler tokens act as intermediate reasoning steps in multi-token computations, yielding measurable performance gains on parallelizable problems. Furthermore, implicit reasoning methods transform explicit, token-level reasoning trajectories into internal reasoning to enhance efficiency or accuracy [6, 7]. For instance, CODI [35] employs a self-distillation to framework to align explicit and implicit reasoning tokens for improved performance. Concurrent to our work, hidden reasoning approaches [11, 34, 36] leverage previous output hidden states as next input embeddings, enabling compact yet informative internal reasoning. Nonetheless, the majority of existing methods require processed traces and extensive training. In contrast, we focus on hybrid latent reasoning through reinforcement learning to exploit the inherent generation capabilities of LLMs.

### 2.2 Reinforcement Learning

Reinforcement learning (RL) is a paradigm where an agent interacts with an environment, receives feedback, and learns to make decisions that maximize cumulative rewards over time [37]. Recently, RL has been introduced to improve language models by learning from implicit human feedback (RLHF) [28]. Such fine-tuning typically employs policy gradient algorithms and their variants like REINFORCE [38]. To reduce variance, actor-critic methods like A2C [26] are proposed to compute a learnt baseline and leverage advantage estimates for better training dynamics. Similarly, proximal policy optimization (PPO) [32] introduces a clipped surrogate objective to bound policy updates, thereby achieving training stability and robustness to hyperparameter choices. Parallel to these approaches, direct preference optimization (DPO) [31] is introduced to directly optimize language models using pairwise human preference comparisons. DPO's simpler variant such as SimPO [25] further mitigates the need of reference models. Despite DPO's efficiency, online RL methods remain preferred for their consistent superior performance [44]. Recently, reinforce leave-one-out (RLOO) [1] proposes REINFORCE-style RL that generates multiple responses and utilizes the mean reward of the other responses as a baseline. Similarly, group relative policy optimization (GRPO) [33] and REINFORCE++ [16] compute baselines from group-level or batch-level reward scores across candidate completions, and thus reduce memory overhead while maintaining accuracy and stability for complex tasks. In this work, we design a novel online RL–driven approach to incentivize hybrid latent reasoning by progressively incorporating hidden states into LLM inputs, thereby providing richer representations for improved reasoning performance.

## 3 Methodology

### 3.1 Hybrid Reasoning with Gating

We first describe our notation and settings for hybrid latent reasoning. For input query $x = [x_1, x_2, \ldots, x_t]$ and its corresponding token embeddings $E = [e_1, e_2, \ldots, e_t]$, we describe the raw hidden states from the LLM output at step $t$ with $\hat{h}_t$, namely:

$$\hat{H} = [\hat{h}_1, \hat{h}_2, \ldots, \hat{h}_t] = \texttt{Transformer}(E), \tag{1}$$

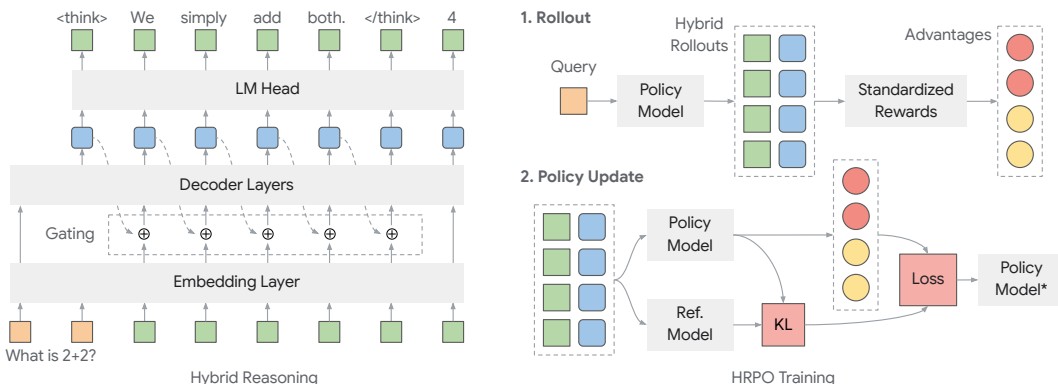

Figure 2: Hybrid reasoning with gating (left) and hybrid reasoning policy optimization (right). During rollouts, the reasoning trajectory is generated hybridly with both discrete tokens and latent features, and for policy update, we compute the HRPO loss using the hybrid rollout buffer to update the model.

in which `Transformer` denotes the transformer model (i.e., decoder layers), $\hat{H}$ represents the final-layer hidden states produced by the `Transformer`. With the LM head (`Head`), the next output token $\hat{x}_{t+1}$ can be sampled from the output distribution over the vocabulary via:

$$\hat{x}_{t+1} \sim \texttt{softmax}(\texttt{Head}(\hat{h}_t)). \tag{2}$$

However, hidden states often lie outside the model's token embedding manifold, which degrades generation quality when fed directly. To avoid this, we project $\hat{h}_t$ back into the embedding space to ensure the inputs conform to the model's learned distribution. Specifically, we use the output probabilities $p_{t+1}$ to compute a weighted interpolation over the vocabulary:

$$h_{t+1} = W_e^T \frac{p_{t+1}}{\|p_{t+1}\|}, \quad \text{with} \quad p_{t+1} = \texttt{softmax}(\frac{\texttt{Head}(\hat{h}_t)}{\tau}), \tag{3}$$

in which $\tau$ is the temperature and $W_e$ denotes the embedding matrix of the LLM. In other words, we compute the next input embedding as a weighted sum of all token embeddings, with weights given by $p_{t+1}$. In addition, $p_{t+1}$ is normalized to preserve the scale and variance of the output vector. This sampling-free mapping ensures differentiability and aligns the projected embedding with the model's native input space, thus leading to improved training dynamics (see Section 4.3).

While interpolated embeddings preserve semantic continuity, directly feeding $h_{t+1}$ as the next token input removes stochasticity and injects noise from irrelevant tokens, causing degraded generation within RL rollouts. As such, we design a hybrid approach for latent reasoning by gradually imposing hidden state representations into the sampled token embeddings with a gating mechanism. Drawing on gated recurrence models [5, 27], we formulate the gating mechanism as:

$$
\begin{aligned}
r_t &= \sigma(W_a \hat{e}_{t+1} + b_a), \\
i_t &= \sigma(W_x \hat{e}_{t+1} + b_x), \\
a_t &= \exp(-c \cdot \texttt{softplus}(\Lambda) \odot r_t), \\
e_{t+1} &= \begin{cases} a_t \odot \hat{e}_{t+1} + \sqrt{1 - a_t^2} \odot (i_t \odot h_{t+1}) & t \in \texttt{think}, \\ \hat{e}_{t+1} & t \notin \texttt{think}, \end{cases}
\end{aligned} \tag{4}
$$

$e_{t+1}$ is the resulting hybrid input for the next step, $\hat{e}_{t+1}$ denotes the embedding of the sampled discrete token $\hat{x}_{t+1}$, whereas $h_{t+1}$ is the projected hidden states as in Equation (3). The gates $r_t$ and $i_t$ leverages sigmoid function $\sigma$ to control the blending, $a_t$ scales $\hat{e}_{t+1}$, $c$ is a fixed scaling constant, and $\Lambda$ is a learnable vector. Note that hybrid reasoning only applies during the reasoning phase (i.e., $t \in \texttt{think}$), while the final answer is still generated via standard autoregressive decoding, as we show in Figure 2 (left). By initializing $a_t \to 1$ (see Section A), the inputs first draw predominantly from the sampled token embeddings, thereby effectively preserving the LLM's generative capabilities. As the training progresses, the value range of $a_t$ converges to an optimum range and thus incorporates informative features from both hidden representations and sampled tokens.

Overall, our hybrid reasoning approach projects hidden states into the embedding space via weighted interpolation. Moreover, the sampling steps preserve stochasticity for effective reinforcement learning. We employ a plug-and-play gating mechanism that initially prioritizes sampled token embeddings while gradually integrating latent signals, providing richer inputs for subsequent reasoning.

### 3.2 Hybrid Reasoning Policy Optimization (HRPO)

Rather than relying on strong supervision, we optimize the policy model via hybrid rollouts using reinforcement learning (RL), fully harnessing LLMs' native reasoning capabilities. Inspired by recent RL advances such as group relative policy optimization (GRPO) [33], we introduce hybrid reasoning policy optimization (HRPO), an efficient RL-driven framework that enable LLMs to fuse discrete tokens with continuous representations for hybrid reasoning.

As illustrated in Figure 2 (right), the proposed HRPO optimizes the policy (parameterized by $\theta$) to maximize the expected reward for input $x$ drawn from dataset $\mathcal{D}$ and the sampled hybrid outputs $y$ (discrete tokens) and $H$ (hidden representations):

$$\max_{\theta} \mathbb{E}_{(x,y)\sim\mathcal{D},(\hat{y},H)\sim\pi_\theta(\cdot|x)}[r(a,y)], \tag{5}$$

where $r$ is a simple outcome-based reward function and $a$ denotes the ground truth answer (i.e., it outputs 1 for correct prediction in $y$ and 0 otherwise). The rewards are computed solely on the discrete tokens within the answer span. To obtain an unbiased, low-variance advantage for hybrid latent reasoning, we generate $g$ hybrid rollouts per input query and compute the advantages by standardizing the rewards within the group (i.e., for the $i$-th response, the advantage is calculated by $\hat{A}_i = \frac{r_i - \mathtt{mean}([r_1, r_2, ..., r_g])}{\mathtt{std}([r_1, r_2, ..., r_g])}$). Consequently, the policy gradients can be estimated with:

$$\nabla_\theta \mathcal{J}_{\mathrm{HRPO}}(\theta) = \mathbb{E}_{x\sim\mathcal{D}, \{(y_i,H_i)\}_{i=1}^g \sim \pi_\theta(\cdot|x)}$$
$$\left[ \frac{1}{g}\sum_{i=1}^g \frac{1}{|y_i|}\sum_{t=1}^{|y_i|} \nabla_\theta \log \pi_\theta(y_{i,t}|x, y_{i,<t}, H_{i,<t})\hat{A}_{i,t} \right] - \beta\nabla_\theta \mathbb{D}_{KL}[\pi_\theta \| \pi_{\mathrm{ref}}], \tag{6}$$

where $\pi_{\mathrm{ref}}$ denotes the reference model and KL-divergence acts as a regularizer, controlled by hyperparameter $\beta$. This objective follows a simple REINFORCE-style formulation, fusing discrete token inputs with continuous hidden representations across the reasoning span via the introduced gating mechanism. The hybrid trajectories that yield higher returns are assigned larger advantage estimates, encouraging policy updates to increase the log probabilities of their subsequent reasoning tokens. For the KL divergence term, we compute log probabilities using solely token IDs for $\pi_{\mathrm{ref}}$, as we find it more effective in preserving training stability. Different from PPO / GRPO objectives, we omit the likelihood ratio and directly use raw log probabilities in Equation (6) because ratio clipping is rarely encountered under our conservative learning schedule. Furthermore, since the hidden representations are directly tied to the parameters $\theta$, each trajectory should only be used for a single gradient update; attempting to reuse it—even with importance sampling—violates the on-policy constraints. As such, our HRPO implementation remains lightweight, strictly on-policy and could be seamlessly combined with further RL optimizations.

In summary, the proposed HRPO framework unifies hybrid latent reasoning under a simple RL objective that fully leverages LLMs' intrinsic reasoning capabilities. During rollouts, the decoding process progressively fuses discrete and continuous representations through a learnable gate, preserving coherence while exploiting hidden states. For policy updates, HRPO derives advantages directly from outcome rewards and performs policy gradient steps with KL regularization. As a result, HRPO incentivizes LLMs to dynamically integrate sampled tokens with latent representations, delivering stable and efficient on-policy hybrid reasoning training without a separate value function.

## 4 Experiments

We evaluate HRPO on both knowledge- and reasoning-intensive tasks: (1) open-domain & multi-hop knowledge-intensive question answering (Knowledge); and (2) science, technology, engineering or mathematics (STEM) benchmarks. The experimental results are reported as follows.

Table 1: Evaluation performance of various larger LLMs and trained models on open-domain and multi-hop QA benchmarks. The table reports exact match scores based on top-3 retrieved documents on five datasets: NQ, TriviaQA, HotpotQA, 2WikiMQA and Bamboogle. The upper block reports results for several RAG baselines using the larger Qwen 2.5 7B LLM, while the lower two blocks evaluate smaller Qwen models (1.5B and 3B) trained with different strategies.

| | NQ | TriviaQA | HotpotQA | 2WikiMQA | Bamboogle | Average |
|---|---|---|---|---|---|---|
| Qwen2.5-7B-Instruct | | | | | | |
| QA | 0.134 | 0.408 | 0.183 | **0.250** | 0.120 | 0.219 |
| CoT | 0.048 | 0.185 | 0.092 | 0.111 | 0.232 | 0.134 |
| IRCoT | 0.224 | 0.478 | 0.133 | 0.149 | 0.224 | 0.242 |
| Search-o1 | 0.151 | 0.443 | 0.187 | 0.176 | **0.296** | 0.251 |
| RAG | **0.349** | **0.585** | **0.299** | 0.235 | 0.208 | **0.335** |
| Qwen2.5-1.5B-Instruct | | | | | | |
| SFT | 0.094 | 0.193 | 0.129 | 0.210 | 0.024 | 0.130 |
| RAG | 0.288 | 0.477 | 0.228 | 0.203 | 0.072 | 0.254 |
| PPO | 0.327 | 0.527 | 0.256 | 0.242 | 0.184 | 0.307 |
| GRPO | 0.293 | 0.480 | 0.202 | 0.213 | 0.120 | 0.261 |
| HRPO (Ours) | **0.364** | **0.553** | **0.273** | **0.276** | **0.216** | **0.337** |
| Qwen2.5-3B-Instruct | | | | | | |
| SFT | 0.249 | 0.292 | 0.186 | 0.248 | 0.112 | 0.217 |
| RAG | 0.348 | 0.544 | 0.255 | 0.226 | 0.080 | 0.291 |
| PPO | 0.356 | 0.563 | 0.304 | 0.293 | 0.240 | 0.351 |
| GRPO | **0.381** | 0.570 | 0.308 | 0.303 | 0.272 | 0.367 |
| HRPO (Ours) | 0.378 | **0.593** | **0.316** | **0.318** | **0.296** | **0.380** |

## 4.1 Evaluation on Knowledge Benchmarks

We first evaluate HRPO on five open-domain and multi-hop question answering (QA) datasets: Natural Questions (NQ), TriviaQA, HotpotQA, 2WikiMultiHopQA (2WikiMQA) and Bamboogle [14, 19, 21, 30, 48]. For each query, we use the E5 embedding model [42] to retrieve the top-3 Wikipedia documents as context (details presented in Section A). Following [18], we merge the NQ and HotpotQA training sets to train HRPO models, and evaluate it on each dataset's evaluation split. The exact match results of HRPO and baselines (including supervised fine-tuning (SFT), retrieval augmented generation (RAG) [22] and RL-based PPO [32] and GRPO [33]) for the 1.5B and 3B Qwen2.5 Instruct models [46] are presented in Table 1. We also include comparisons to several QA and RAG baselines using the larger Qwen2.5-7B-Instruct as backbone, including: direct inference (QA), chain-of-thought (CoT) [43], interleaving retrieval with CoT (IRCoT) [41], Search-o1 [23] and RAG [22]. For each block in Table 1, we mark the best performance in bold for clarity.

Across all knowledge benchmarks, HRPO delivers the strongest exact match (EM) scores with smaller Qwen models and rivals the much larger 7B baselines. In particular, we observe: (1) HRPO reaches 0.380 EM with Qwen2.5-3B, outperforming the strongest 7B RAG baseline by 4.5%. Similarly, HRPO with the smaller 1.5B backbone scores an average of 0.337, achieving consistent gains and surpassing PPO by 3.0%. (2) HRPO consistently outperforms other RL-based methods. For example, HRPO with both the 1.5B and 3B backbones surpasses the strongest RL baseline by 3.0% and 1.3% respectively; the only dataset both models perform similarly is NQ. (3) Interestingly, GRPO underperforms PPO by 4.6% on the 1.5B backbone but outperforms it by 1.6% on the 3B model, likely a consequence of sparser rewards and limited sampled trajectories with a smaller model. (4) RL-based methods perform on par with the best-performing RAG baseline, with HRPO delivering the largest performance gains—particularly on terse, incomplete queries (NQ) and multi-hop questions (2WikiMQA)—while yielding modest improvements on one-hop datasets like TriviaQA. Overall, these results demonstrate that combining retrieval augmentation with hybrid latent reasoning yields state-of-the-art knowledge performance under computation constraints, establishing HRPO as a competitive alternative to both RL-based learning methods and larger retrieval augmented LLMs.

Table 2: Evaluation performance of various larger LLMs and trained models on STEM benchmarks. The table presents accuracy scores on five datasets: GSM8k, MATH, MATH500, MMLU-ST and ARC-C. The upper block reports results for several few-shot baseline LLMs $\geq$ 7B, while the lower two blocks evaluate smaller Qwen models (1.5B and 3B) trained with different strategies.

| | GSM8k | MATH | MATH500 | MMLU-ST | ARC-C | Average |
|---|---|---|---|---|---|---|
| Larger LLMs (Size $\geq$ 7B) | | | | | | |
| DeepSeekMath-7B | 0.642 | 0.362 | 0.346 | 0.565 | 0.678 | 0.519 |
| Gemma-2-9B | 0.707 | 0.377 | 0.364 | 0.651 | 0.682 | 0.556 |
| Qwen2.5-7B | **0.854** | **0.498** | **0.464** | **0.723** | 0.637 | 0.635 |
| MAmmoTH2-7B | 0.684 | 0.367 | 0.396 | 0.624 | 0.817 | 0.578 |
| MAmmoTH2-8B | 0.704 | 0.358 | 0.732 | 0.642 | **0.822** | **0.652** |
| Qwen2.5-1.5B-Instruct | | | | | | |
| SFT | 0.560 | 0.300 | 0.302 | 0.403 | 0.602 | 0.433 |
| Distilled CoT | 0.706 | 0.503 | - | - | - | - |
| PPO | 0.694 | 0.507 | 0.518 | 0.566 | 0.715 | 0.600 |
| GRPO | 0.711 | 0.502 | 0.524 | 0.562 | 0.737 | 0.607 |
| HRPO (Ours) | **0.720** | **0.518** | **0.536** | **0.569** | **0.742** | **0.617** |
| Qwen2.5-3B-Instruct | | | | | | |
| SFT | 0.670 | 0.348 | 0.360 | 0.454 | 0.474 | 0.461 |
| Distilled CoT | 0.799 | 0.575 | - | - | - | - |
| PPO | 0.819 | 0.597 | 0.604 | 0.582 | 0.811 | 0.682 |
| GRPO | 0.834 | 0.602 | 0.604 | **0.601** | 0.814 | 0.691 |
| HRPO (Ours) | **0.845** | **0.613** | **0.630** | 0.590 | **0.820** | **0.700** |

## 4.2 Evaluation on STEM Benchmarks

We also evaluate the performance of the proposed HRPO on the reasoning-intensive STEM datasets: GSM8k, MATH, MATH500, MMLU-STEM (MMLU-ST) and ARC-Challenge (ARC-C) [4, 13, 24, 12, 3]. Table 2 reports the performance of HRPO alongside fine-tuned baselines (SFT, SFT with distilled CoT from QwQ [40]) and RL baselines (PPO [32] and GRPO [33]) on the Qwen 2.5 1.5B and 3B Instruct models [46]. In addition, we select several larger LLMs ($\geq$ 7B in size) using few-shot CoT for comparison [46, 33, 49]. For GSM8k, we train on the training split, and for MATH and MATH500, we train on the MATH training split. For MMLU-ST and ARC-C, we train on the merged auxiliary MMLU and ARC-C training sets. Distilled CoT is only available for GSM8k and MATH due to dataset size constraints. We also highlight the best scores in each block in bold.

Across the five STEM benchmarks, HRPO delivers the strongest results with compact Qwen backbones and could match the performance of much larger LLMs. Our key observations are: (1) SFT underperforms compared to distilled CoT and RL-based methods, suggesting the efficacy of RL with verifiable rewards on reasoning-intensive tasks. (2) With the 3B backbone, HRPO achieves an average accuracy of 0.700, matching the best 7B baseline on four of the datasets. Even the 1.5B HRPO averages at 0.617, outperforming the 7B leader on MATH by 2.0%. (3) At 1.5B, HRPO improves on the strongest alternative GRPO with notable boosts on MATH and MATH500 (1.6% and 1.2%), whereas the average gain narrows at 3B, implying that HRPO is more beneficial for smaller models. (4) HRPO registers the highest accuracies recorded for sub-7B models on MATH (0.613) and MATH500 (0.630), demonstrating the value of RL-based hybrid reasoning on challenging benchmarks. Taken together, these results show that hybrid latent reasoning unlocks the power of much larger LLMs in compact backbones, proving the effectiveness of the proposed HRPO.

## 4.3 Analysis of HRPO

**Different Strategies for Latent Reasoning.** We compare different strategies to compute latent representations. Specifically, we use three methods to integrate hidden states into RL and train the 1.5B Qwen model on the MATH dataset. These variants are: (1) hidden states, which use the final layer hidden states as the next input; (2) interpolation, which employs interpolated embeddings

as defined in Equation (3); and (3) HRPO, our hybrid latent reasoning in Equation (4). We visualize the exponential moving average (EMA) of rewards along with the GRPO baseline in Figure 3. Due to the mismatch between hidden states and embeddings, using hidden states degrades generation and yields nonsensical rollouts with zero reward. Although interpolation performs similar to HRPO for the first few hundred steps, the rewards eventually collapse and only slowly recover, likely because interpolation introduces excessive noise. We also provide a direct comparison between HRPO and latent reasoning methods in Section B. Overall, our

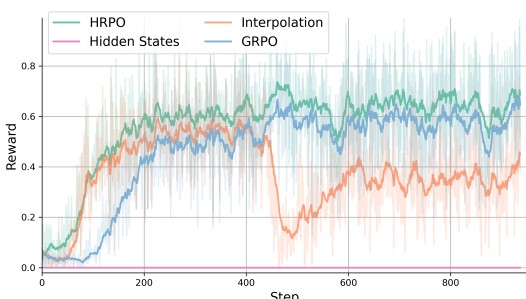

Figure 3: Reward on MATH for Qwen-2.5-1.5B using different latent reasoning strategies.

approach achieves superior training dynamics with faster convergence while maintaining stability comparable to GRPO, highlighting the efficacy of our hybrid design choice in HRPO.

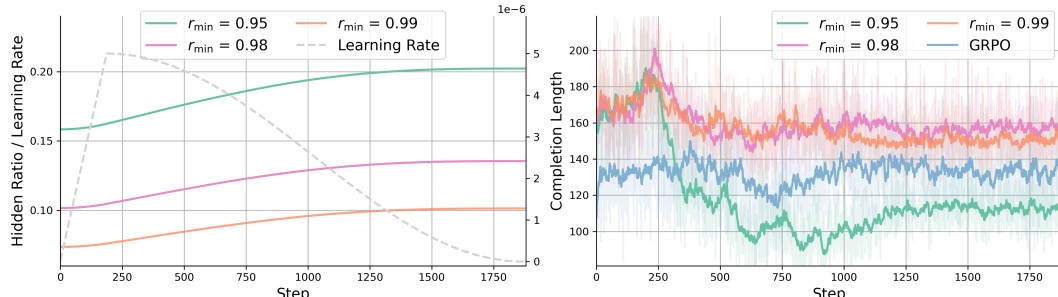

Figure 4: Hidden ratio with varying $r_{\min}$ in $\exp(-c \cdot \texttt{softplus}(\Lambda))$ and learning rate. We visualize the hidden ratio and completion length for training runs with $r_{\min}$ from $[0.95, 0.98, 0.99]$.

**Ratio of Latent Representations.** We track how the balance between discrete tokens and continuous latent representations shifts as LLMs learn to reason hybridly. Here, we train Qwen 1.5B on the knowledge task and visualize both the mean hidden ratios (i.e., $\sqrt{1 - a_t^2}$) and completion lengths (along with GRPO) in Figure 4. Across all runs, the hidden ratio increases steadily, even as the learning rate tapers off toward the end of training under a cosine schedule. In addition, completion lengths increase during the initial phase and later decline across all methods, with the drops most significant in HRPO. Furthermore, setting $r_{\min} = 0.95$ leads to an interesting behavior where completion lengths substantially decrease—an effect not seen in the other variants[2]. This may be because the hidden representations effectively capture historical context, thereby shortening completions while maintaining or even improving performance (see Table 3). As such, hybrid latent reasoning could be particularly effective when leveraging contextual information for reasoning.

Table 3: Impact of $\Lambda$-initialization on HRPO's performance across knowledge and STEM tasks.

| Init Range | Knowledge | | | | | |
|---|---|---|---|---|---|---|
| | NQ | TriviaQA | HotpotQA | 2WikiMQA | Bamboogle | Average |
| [0.95 - 0.999] | **0.364** | **0.553** | **0.273** | 0.264 | 0.184 | 0.328 |
| [0.98 - 0.999] | 0.336 | 0.553 | 0.263 | **0.276** | **0.216** | **0.329** |
| [0.99 - 0.999] | 0.336 | 0.534 | 0.258 | 0.275 | 0.216 | 0.324 |

| Init Range | STEM | | | | | |
|---|---|---|---|---|---|---|
| | GSM8k | MATH | MATH500 | MMLU-ST | ARC-C | Average |
| [0.95 - 0.999] | 0.705 | 0.516 | **0.536** | **0.569** | 0.735 | 0.612 |
| [0.98 - 0.999] | 0.703 | 0.509 | 0.532 | 0.563 | 0.732 | 0.608 |
| [0.99 - 0.999] | **0.720** | **0.518** | 0.526 | 0.567 | **0.742** | **0.614** |

[2]$r_{\min}$ is used to initialize $\Lambda$ such that $\exp(-c \cdot \texttt{softplus}(\Lambda))$ is drawn uniformly from $[r_{\min}, 0.999]$.

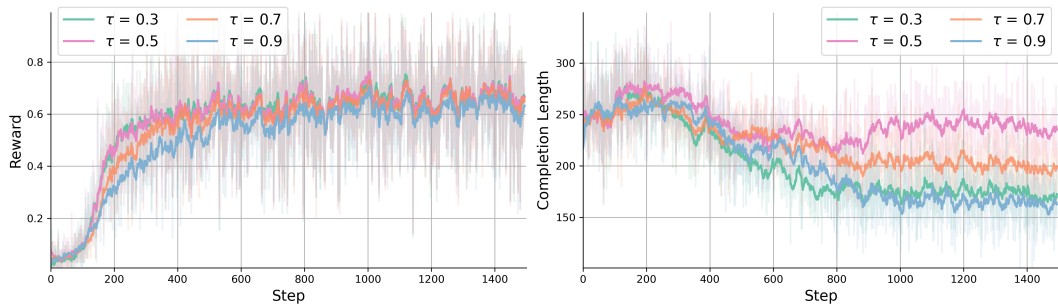

Figure 5: Sensitivity analysis for temperature $\tau$ in Equation (3). We visualize the reward and completion length for training runs with different temperature selected from $[0.3, 0.5, 0.7, 0.9]$.

**Initialization of $\Lambda$ for Gating.** Beyond hidden ratio, we examine how the initialization of $\Lambda$—which control the balance between latent features and token embeddings—affects HRPO performance. Specifically, we initialize $\exp(-c \cdot \texttt{softplus}(\Lambda))$ from $[r_{\min}, 0.999]$ and report the results on Qwen 1.5B in Table 3, where lowering $r_{\min}$ yields a higher initial hidden ratio. For the knowledge domain, performance improves as $r_{\min}$ decreases: the best average performance occurs at $r_{\min} = 0.98$, and most individual datasets peak at $r_{\min} = 0.95$. In contrast, the STEM benchmarks display a bimodal trend: performance rises when $r_{\min}$ is either lower or higher, but drops for the intermediate range $[0.98, 0.999]$. This pattern implies that the model profits from emphasizing either explicit token trajectories or latent representations, whereas a mid-level mix is sub-optimal. In summary, our results show that knowledge tasks benefit from lower $r_{\min}$, whereas optimal performance for STEM tasks arises from leaning toward either explicit token trajectories or latent representations.

**Sensitivity of $\tau$ on Hybrid Reasoning.** We further investigate the impact of temperature $\tau$ on HRPO: lower $\tau$ values reduce noise but overemphasize top tokens, whereas larger $\tau$ spreads probability mass across more tokens. We explore $\tau \in \{0.3, 0.5, 0.7, 0.9\}$ and present the rewards and completion lengths of the 1.5B Qwen model on MMLU in Figure 5. The left panel indicates that $\tau = 0.3$ and $\tau = 0.5$ converge faster and reach the highest reward plateau, outperforming higher values ($\tau \geq 0.7$) and showing the benefits of a smaller $\tau$. Interestingly, the right panel reveals that both smaller and larger $\tau$ values shorten completion lengths, while $\tau = 0.5$ and $\tau = 0.7$ maintain longer generations. This may be because lower $\tau$ sharpens token distribution, yielding a confident latent vector that lets HRPO finish quickly. In contrast, higher $\tau$ flattens the distribution and enhances informativeness, prompting the policy to extract answers in shorter rollouts. Overall, we find HRPO to be robust across varying $\tau$ selections, only completion length varies noticeably. Further analysis is in Section B.

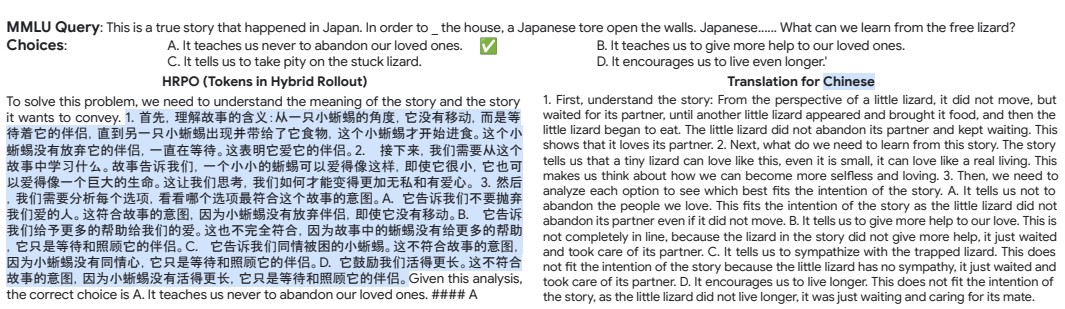

Figure 6: Example cross-lingual reasoning (English-Chinese) and its translation for HRPO.

**Hybrid Latent Reasoning Patterns.** Finally, we highlight several intriguing reasoning patterns that emerge from HRPO. First, the hybrid outputs show readable trajectories by interpreting the tokens even without any CoT supervision. Second, HRPO exhibits cross-lingual patterns in some completions, fluidly integrating tokens from different languages, suggesting that latent representations can generalize across linguistic boundaries (see Figure 6). Moreover, the hybrid reasoning process often delivers compact yet accurate responses to simple or factual queries, where the model requires fewer decoding steps thanks to the richer context encoded in the hidden representations. These

emergent patterns indicate that hybrid latent reasoning can improve both interpretability and efficiency over existing latent reasoning approaches. Further qualitative examples can be found in Section C.

## 5 Conclusion

In this work, we propose hybrid reasoning policy optimization (HRPO), a novel reinforcement learning (RL) framework that unifies discrete token sampling with continuous latent representations through a learnable gating mechanism. By gradually incorporating hidden features into sampled token embeddings, HRPO incentivizes LLMs to refine their reasoning strategies hybridly. Extensive evaluations on knowledge and STEM benchmarks demonstrate that HRPO outperforms both SFT and RL baselines, achieving consistent gains across diverse scenarios. Moreover, our analysis reveals that HRPO not only ensures stable hybrid latent reasoning but also triggers intriguing reasoning patterns, showing its potential in reasoning-intensive settings and providing insights for RL-based continuous space learning. While promising, we recognize that HRPO introduces additional computation overhead, the on-policy design may reduce large-scale training efficiency, and its continuous representations can be less transparent. Therefore, future work will aim to address these limitations by exploring simpler designs, off-policy extensions and advanced latent reasoning techniques to improve both the interpretability and efficiency of HRPO.

## 6 Acknowledgments

This research is supported in part by the National Science Foundation under Grant No. CNS-2427070, IIS-2331069, IIS-2202481, IIS-2130263, CNS-2131622. The views and conclusions contained in this document are those of the authors and should not be interpreted as representing the official policies, either expressed or implied, of the U.S. Government. The U.S. Government is authorized to reproduce and distribute reprints for Government purposes notwithstanding any copyright notation here on.

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

## A    Implementation

For hybrid latent reasoning, our plug-and-play component is by design compatible with any LLM architecture. We initialize its linear layers with a uniform distribution from $[-1/\sqrt{|H|}, 1/\sqrt{|H|}]$, where $|H|$ denotes the hidden state dimension. The gating parameter $\Lambda$ is selected such that the quantity $a^c = \exp(-c \cdot \texttt{softplus}(\Lambda))$ is drawn uniformly from $[r_{\min}, 0.999]$, with the scalar constant fixed at $c = 8$ [5]. Tuning $r_{\min}$ adjusts the initial fraction of hidden states involved in hybrid reasoning; a larger value increases the proportion of sampled token embeddings and can be helpful for enhancing generation quality during the initial training phase. Similarly, the temperature hyperparameter $\tau$ in Equation (3) can be tuned for optimal task performance, although HRPO remains robust across a wide range of $\tau$ values. To efficiently train the LLMs with HRPO, we patch the models with optimized kernel implementations[3] and employ low-rank adaptation (LoRA) [15]. The default choice of hyperparameters are reported in Table 4 for HRPO experiments.

Table 4: Experiment hyperparameter settings.

| | |
|---|---|
| Algorithm | HRPO |
| Epochs | 1 |
| Optimizer | AdamW 8bit |
| Optimizer Momentum | $\beta_1$, $\beta_2$ = 0.9, 0.99 |
| Weight Decay | 0.1 |
| Learning Rate | 5e-6 |
| Learning Rate (Linear in Equation (4)) | 1e-4 |
| Learning Rate ($\Lambda$ in Equation (4)) | 1e-3 |
| HRPO $\beta$ | 0.005 |
| Max Gradient Norm | 0.1 |
| Gradient Accumulation Step | 4 |
| Group size $g$ in HRPO | 4 / 8 |
| Total Train Batch Size | 32 / 64 |
| LR Scheduler | Cosine with Warmup |
| Warmup Ratio | 0.1 |
| Precision (WA) | BF16-mixed |
| LoRA Modules | query, key, value, dense |
| LoRA Rank | 32 |
| LoRA $\alpha$ | 64 |

The hyperparameters are selected empirically to balance efficiency and performance, and thanks to HRPO's lightweight design and additional optimizations, our framework can run on a single GPU across diverse tasks. Additionally, we apply a larger weight-decay coefficient to (1) enhance HRPO training stability and (2) encourage the gating towards incorporating more latent representations (since smaller positive $\Lambda$ values increase the hidden ratio $\sqrt{1 - a_t^2}$). For simpler knowledge tasks and GSM8k, we fix the HRPO group size at 4, which already delivers strong performance. For more challenging benchmarks, namely MATH, MATH500, MMLU-ST and ARC-C, we instead generate 8 hybrid completions for each query. As for prompt and completion lengths, we select them empirically based on our observations, and the selected values are summarized in Table 5.

Table 5: Experiment prompt / completion lengths.

| | |
|---|---|
| Prompt / Completion Length for Knowledge Tasks | 2048 / 512 |
| Prompt / Completion Length for GSM8k | 512 / 512 |
| Prompt / Completion Length for MATH & MATH500 | 512 / 1024 |
| Prompt / Completion Length for MMLU-ST & ARC-C | 512 / 512 |

For both training and evaluation, we build each prompt by prepending a system message that directs the LLM to perform step-by-step internal reasoning before generating its final answer. The user query is then appended, and the entire input is formatted with the model chat template. Different from

---

[3]https://github.com/unslothai/unsloth

prior work [10, 18], we adopt the minimalist delimiter #### to separate the model's hybrid reasoning span from its final answer. This is because the delimiter tokenizes as a single unit, adding no length overhead while providing a clear signal to switch from hybrid latent reasoning to autoregressive answer generation. We also penalize repeated occurrences of the delimiter within the completion (by assigning 0 reward regardless answer correctness) to prevent the model from early termination of hybrid reasoning. We illustrate full prompts for different type of tasks, showing the system message and example queries in Figure 7, Figure 8 and Figure 9, respectively.

---

**Example Prompt for Knowledge Tasks**

```
<|im_start|>system
A conversation between User and Assistant.  The user asks a question,
and the assistant solves it.  The assistant first thinks about the
reasoning process in the mind and then provides the user with the
answer.  The final answer is provided after the #### tag, i.e.,
{reasoning process} #### {answer}.<|im_end|>
<|im_start|>user
Context (which may or may not be relevant):
Clyde River (New South Wales)::::Clyde River (New South Wales) The...
Barwon River (New South Wales)::::River and Weir River (part of...
Taponga River::::Taponga River The Taponga River, an inland...

Question:  What direction does the river that Austrolebias bellotti
are found in flow?<|im_end|>
<|im_start|>assistant
```

Figure 7: Example prompt for knowledge tasks, contexts are partially omitted due to space constraints.

---

**Example Prompt for GSM8k / MATH / MATH500**

```
<|im_start|>system
A conversation between User and Assistant.  The user asks a question,
and the assistant solves it.  The assistant first thinks about the
reasoning process in the mind and then provides the user with the
answer.  The final answer is provided after the #### tag, i.e.,
{reasoning process} #### {answer}.<|im_end|>
<|im_start|>user
Natalia sold clips to 48 of her friends in April, and then she
sold half as many clips in May.  How many clips did Natalia sell
altogether in April and May?<|im_end|>
<|im_start|>assistant
```

Figure 8: Example prompt for GSM8k / MATH / MATH500 in HRPO.

---

For each question in our knowledge-intensive QA setup, we embed the query with E5 embedding model [42]. The entire English Wikipedia 2020 dump is pre-encoded with the same model, after which we perform approximate nearest neighbor (ANN) search and select the three highest-scoring documents. These top-3 passages are concatenated to form the external context fed to the LLM, as illustrated in Figure 7. In our evaluation, we generate tokens using greedy decoding and compute latent representations according to Equation (3), thereby ensuring the reproducibility of our results. For outcome-based reward and evaluation settings on knowledge tasks, we report exact match scores on val / test splits following [50, 51, 18]. For mathematical (GSM8k, MATH and MATH500) and multiple-choice datasets (MMLU-ST and ARC-C), we follow [49] for post-processing and scoring.

```
Example Prompt for MMLU-ST / ARC-C

<|im_start|>system
A conversation between User and Assistant.  The user asks a question,
and the assistant solves it.  The assistant first thinks about the
reasoning process in the mind and then provides the user with the
answer.  The final answer is provided after the #### tag, i.e.,
{reasoning process} #### {answer}.<|im_end|>
<|im_start|>user
Question:  Two people are pushing a car.  One person is pushing with
a force of 450 N and the other person is pushing with a force of 300
N. What information is needed to determine the net force applied to
the car by the people?

Options:
A. the direction of the road
B. the direction of the forces
C. the weight of the two people
D. the weight of the automobile<|im_end|>
<|im_start|>assistant
```

Figure 9: Example prompt for MMLU-ST / ARC-C in HRPO.

# B   Additional Results

**Comparison to Latent Reasoning Methods.** In addition to strong RL methods such as PPO and GRPO in our main experiments, we also benchmark the proposed HRPO against additional latent reasoning baselines. Specifically, we evaluate HRPO, Coconut and CODI on the GSM8K and MATH reasoning datasets, all using the 1.5B Qwen backbone. For Coconut, we train with its augmented CoT data (no MATH split is available), whereas for CODI we adopt the original datasets' CoT trajectories. The results are reported in Table 6. We observe: (1) HRPO achieves the best accuracy on both datasets, with 9.42% and 23.63% respective gains over the best performing latent reasoning baseline CODI. (2) Even compared to distilled CoT from a significantly larger model QwQ, HRPO still scores consistent improvements on both datasets, showing the effectiveness of our hybrid latent reasoning. (3) Coconut lags behind on GSM8k, indicating limitations of latent reasoning by compressing CoT tokens, whereas CODI improves substantially with CoT SFT but still trails Distilled CoT and HRPO. Overall, HRPO achieves the best performance against all baselines, demonstrating its consistent advantages over CoT distillation and prior latent reasoning methods.

Table 6: Performance comparison of HRPO against alternative latent reasoning methods and distilled CoT baseline.

|  | Coconut | | CODI | | Distilled CoT | | HRPO | |
| --- | --- | --- | --- | --- | --- | --- | --- | --- |
|  | GSM8k | MATH | GSM8k | MATH | GSM8k | MATH | GSM8k | MATH |
| Accuracy | 0.315 | - | 0.658 | 0.419 | 0.706 | 0.503 | **0.720** | **0.518** |

**Sensitivity Analysis for $\Lambda$ and $\tau$.** In addition to the results reported in Table 3, we further present the performance of various $\Lambda$ initializations on the Qwen 3B model, as shown in Table 7. Our observations echo the same trends on the 1.5B backbone: a smaller initial $r_{min}$ consistently benefits both knowledge and STEM tasks. Moreover, performance peaks when $r_{min}$ is selected either lower or higher, and drops slightly within the intermediate range of $[0.98, 0.999]$. We also examine the sensitivity of the $\tau$ hyperparameter used to construct the interpolated embeddings and present the corresponding results for both backbone models in Table 8. The training rewards and completion lengths for GSM8k, MATH and the knowledge tasks are shown in Figure 10, Figure 11 and Figure 12. We note that choosing $\tau$ in the range of $0.5 - 0.7$ offers a reliable balance of efficiency and accuracy, as the performance often peaks around this interval for both backbone models. Overall, we find that

Table 7: Impact of $\Lambda$-initialization on HRPO's performance for the Qwen 3B backbone.

| Init Range | Knowledge | | | | | |
|---|---|---|---|---|---|---|
| | NQ | TriviaQA | HotpotQA | 2WikiMQA | Bamboogle | Average |
| [0.95 - 0.999] | **0.845** | **0.613** | 0.622 | 0.576 | **0.820** | 0.695 |
| [0.98 - 0.999] | 0.842 | 0.600 | 0.614 | 0.585 | 0.813 | 0.691 |
| [0.99 - 0.999] | 0.838 | 0.606 | **0.630** | **0.590** | 0.817 | **0.696** |

| Init Range | STEM | | | | | |
|---|---|---|---|---|---|---|
| | GSM8k | MATH | MATH500 | MMLU-ST | ARC-C | Average |
| [0.95 - 0.999] | 0.367 | **0.593** | **0.316** | 0.311 | **0.296** | **0.377** |
| [0.98 - 0.999] | **0.378** | 0.588 | 0.311 | 0.298 | 0.296 | 0.374 |
| [0.99 - 0.999] | 0.375 | 0.584 | 0.309 | **0.318** | 0.288 | 0.375 |

HRPO benefits from a smaller initial $r_{\min}$, which outperforms larger $r_{\min}$ settings and highlights the value of latent representations for complex reasoning. Moreover, HRPO is robust to the choice of $\tau$, where the performance scores remain stable with only minor fluctuations at the extremes.

Table 8: Impact of $\tau$ on HRPO's performance for both backbone models.

| Model | Qwen2.5-1.5B | | | | Qwen2.5-3B | | | |
|---|---|---|---|---|---|---|---|---|
| $\tau$ | **0.3** | **0.5** | **0.7** | **0.9** | **0.3** | **0.5** | **0.7** | **0.9** |
| GSM8k | 0.717 | **0.720** | 0.705 | 0.694 | 0.842 | 0.841 | **0.845** | 0.833 |
| MATH | **0.518** | 0.516 | 0.507 | 0.514 | 0.597 | 0.606 | **0.613** | 0.599 |
| MATH500 | 0.522 | **0.536** | 0.532 | 0.524 | 0.622 | 0.614 | 0.622 | **0.630** |
| MMLUST | 0.561 | **0.569** | 0.559 | 0.567 | 0.577 | **0.590** | 0.574 | 0.580 |
| ARC-C | 0.735 | 0.741 | **0.742** | 0.724 | **0.820** | 0.817 | 0.809 | 0.808 |
| NQ | 0.320 | 0.336 | 0.317 | **0.364** | **0.378** | 0.375 | 0.373 | 0.363 |
| TQ | 0.524 | 0.534 | **0.553** | 0.553 | 0.588 | **0.593** | 0.578 | 0.578 |
| HotpotQA | 0.263 | 0.260 | 0.252 | **0.273** | 0.311 | **0.316** | 0.309 | 0.306 |
| 2Wiki | **0.276** | 0.272 | 0.264 | 0.244 | **0.318** | 0.311 | 0.297 | 0.293 |
| Bamboogle | 0.216 | 0.216 | **0.216** | 0.176 | 0.296 | 0.288 | **0.296** | 0.280 |

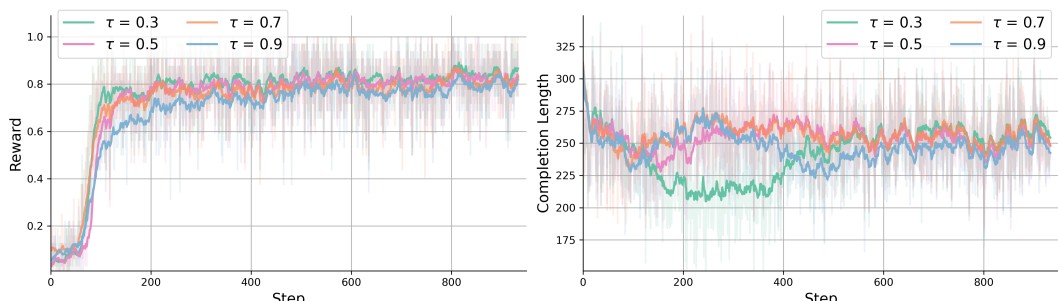

Figure 10: Reward and completion length for training runs with different temperature values on GSM8k using the Qwen 1.5B backbone.

**Additional Analysis for $\Lambda$ Initialization.** We further provide an expanded analysis of how varying $r_{\min}$ in the initialization of $\Lambda$ affects training dynamics with the larger Qwen 3B backbone. Figures Figure 13, Figure 14, Figure 15 and Figure 16 plot the reward and completion length curves for the knowledge tasks, GSM8k, MATH and MMLU-ST / ARC-C respectively. Overall, our findings here echo the observations in Section 4.3: different $r_{\min}$ values exhibit similarly high training stability and preserve the LLM's generative capabilities, but selecting a smaller $r_{\min}$ (i.e., a larger initial hidden ratio) generally accelerates convergence and shortens generated completions. Nevertheless, these benefits are less pronounced for the 3B backbone than for the 1.5B counterpart, which we attribute to the fewer update steps and trainable parameters in HRPO. In summary, our analysis shows

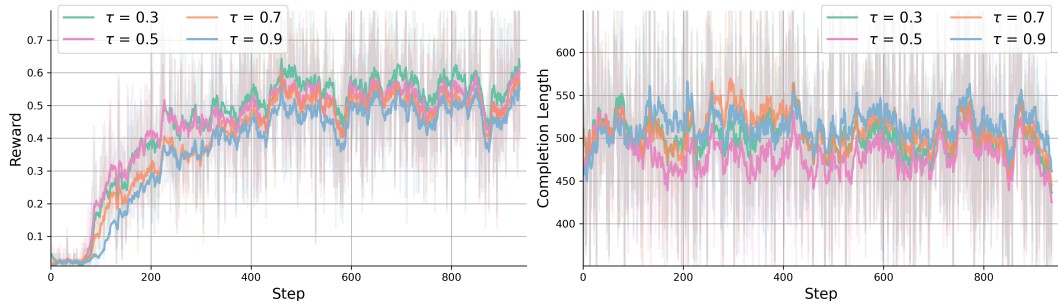

Figure 11: Reward and completion length for training runs with different temperature values on MATH using the Qwen 1.5B backbone.

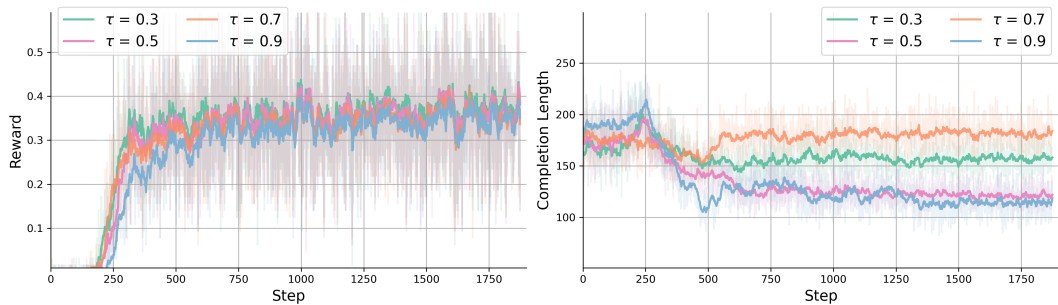

Figure 12: Reward and completion length for training runs with different temperature values on knowledge tasks using the Qwen 1.5B backbone.

that HRPO preserves stable training dynamics and effectively leverages LLMs' intrinsic reasoning patterns across $r_{\min}$ values; moreover, choosing a smaller $r_{\min}$ further enhances convergence and yields shorter generated sequences, which can be especially beneficial for smaller-scale LLMs.

**Statistical Significance Analysis on the Improvements of HRPO.** In our main experiments, we follow the standard practice of using greedy decoding for pass@1 evaluation, ensuring our results are easy to evaluate and reproducible. To evaluate the significance of the performance gains of HRPO, we conduct additional sampling-based evaluations on the STEM tasks, which exhibit greater variance compared to greedy decoding. Averaged results are presented in Table 9, with statistically significant outcomes (paired t-test, $p < 0.05$) highlighted in bold. These results show that HRPO consistently outperforms PPO and GRPO across both backbones on all benchmark datasets. For the 1.5B backbone, t-tests confirm these gains are statistically significant in three out of five tasks. The improvements are even more pronounced with the 3B model, which achieves an average gain of +1.4% and shows statistical significance in four out of five comparisons. These findings demonstrate that our hybrid-RL framework, HRPO, not only delivers reliable performance gains over established baselines but also does so with high statistical confidence across the majority of STEM tasks.

Table 9: Significance test on HRPO's performance improvements.

|  | Qwen2.5-1.5B | | | | |
| --- | --- | --- | --- | --- | --- |
|  | GSM8k | MATH | MATH500 | MMLU-ST | ARC-C |
| PPO | 0.701 | 0.505 | 0.511 | 0.551 | 0.716 |
| GRPO | 0.710 | 0.510 | 0.512 | 0.554 | 0.722 |
| HRPO | 0.712 | **0.515** | 0.517 | **0.565** | **0.731** |
|  | Qwen2.5-3B | | | | |
|  | GSM8k | MATH | MATH500 | MMLU-ST | ARC-C |
| PPO | 0.825 | 0.597 | 0.600 | 0.574 | 0.802 |
| GRPO | 0.827 | 0.595 | 0.599 | 0.577 | 0.808 |
| HRPO | **0.838** | **0.606** | **0.609** | **0.585** | **0.815** |

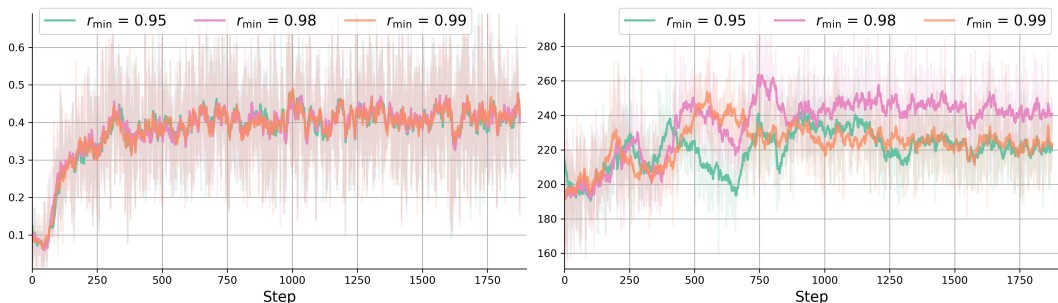

Figure 13: Reward and completion length for training runs with varying initial $r_{\min}$ on knowledge tasks using the Qwen 3B backbone.

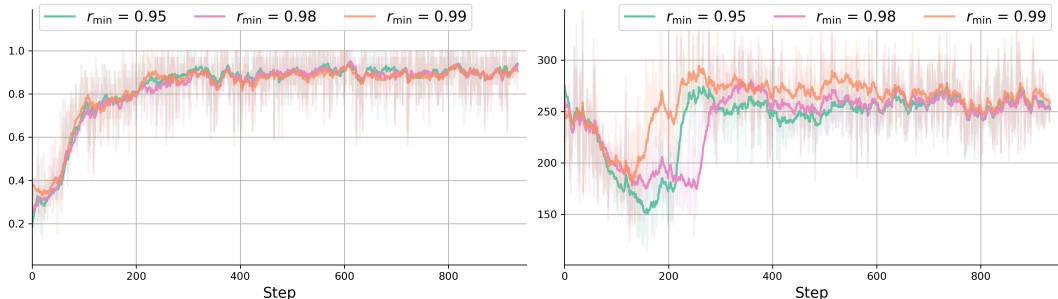

Figure 14: Reward and completion length for training runs with varying initial $r_{\min}$ on GSM8k using the Qwen 3B backbone.

## C   Qualitative Analysis

To further highlight HRPO's reasoning patterns, we present additional qualitative examples. Each example provides the reasoning trace by decoding the sampled tokens from the hybrid reasoning process, and we include both successful and erroneous cases across different tasks in the following. The correct examples are provided in Figure 17, Figure 18, Figure 19, Figure 20, Figure 21, where as the mistakes are provided in Figure 22, Figure 23, Figure 24, Figure 25, Figure 26, we show the raw strings and omit the options / contexts in the examples due to space constraints.

From these examples, we identify four reasoning patterns that can lead to correct answers: (1) Purely English reasoning with coherent trajectories (Figs. Figure 17 and Figure 18), a pattern commonly observed in LLM reasoning outputs. (2) Predominantly English reasoning punctuated by rare tokens (e.g., %n rather than \n), as shown in Figure 19. (3) Cross-lingual reasoning that interweaves multiple languages (English and Chinese in Figure 20). (4) Reasoning with many uncommon tokens and atypical steps, yet still arriving at the correct answer (Figure 21). These latter three patterns are rarely observed in standard reasoning LLMs but are more prevalent in HRPO trained models, demonstrating that HRPO can enhance reasoning by leveraging LLMs' intrinsic generative capabilities across different languages and token types, thereby delivering improvements across diverse scenarios.

As for reasoning errors, we also identify several common patterns: (1) Cross-lingual mistakes arising from limited parametric or contextual knowledge, as in Figure 22 and Figure 23. (2) Correct answers that violate the predefined format and thus receive a zero score (Figure 24). (3) Repetitive loops that continue until the response hits the maximum completion length (Figure 25). (4) Cross-lingual reasoning that is nonetheless truncated by the length limit (Figure 26). Overall, these patterns indicate that, while HRPO effectively integrates discrete and latent representations in its internal reasoning process, it may be further enhanced through refined output formatting (e.g., potentially with a format reward), extended optimization schedules with conservative learning, increased model parameters, and longer context / generation capabilities, pointing to promising directions for future research.

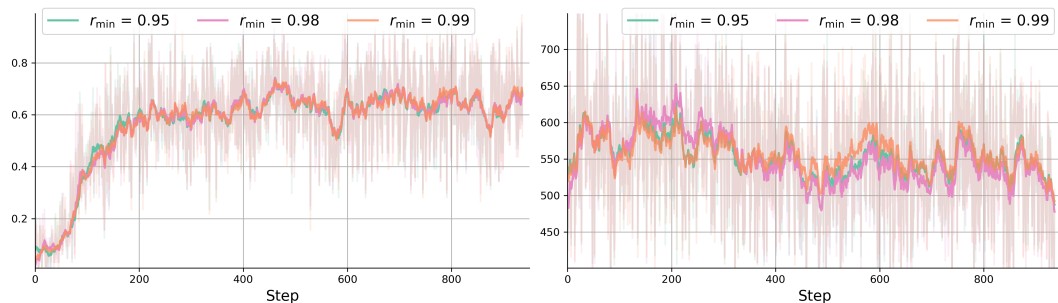

Figure 15: Reward and completion length for training runs with varying initial $r_{\min}$ on MATH using the Qwen 3B backbone.

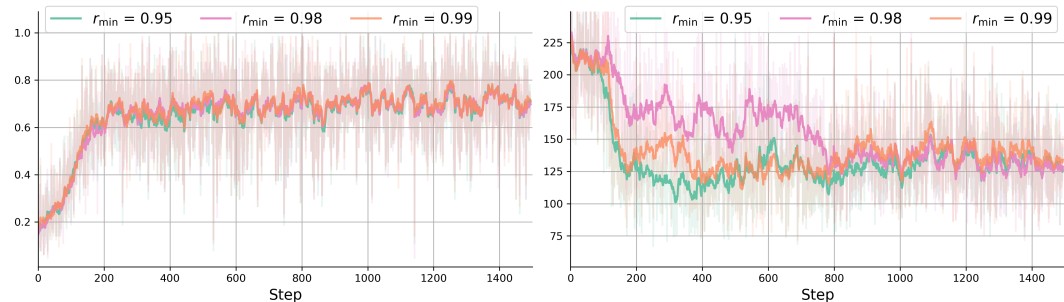

Figure 16: Reward and completion length for training runs with varying initial $r_{\min}$ on MMLU-ST / ARC-C using the Qwen 3B backbone.

**Query**: Which film has the director who died first, The People Vs. Larry Flynt or Manila In The Claws Of Light?

**HRPO (Tokens in Hybrid Rollout):** To determine which film has the director who died first, we need to identify the directors of both films and find their dates of death.\n\nThe People vs. Larry Flynt:\nDirector: Miloš Forman\nMiloš Forman died on August 19, 2012.\n\nManila in the Claws of Light:\nDirector: Lino Brocka\nLino Brocka died on June 1, 1973.\n\nComparing the dates:\nMiloš Forman died on August 19, 2012\nLino Brocka died on June 1, 1973\n\nLino Brocka died first.\n\nTherefore, the film with the director who died first is Manila in the Claws of Light.\n\nThe answer is: Manila in the Claws of Light #### Manila in the Claws of Light ✅

Figure 17: Correct reasoning example 1 in HRPO.

**Query**: Which religion uses a domed structure called a Stupa as a shrine?

**HRPO (Tokens in Hybrid Rollout):** To determine which religion uses a domed structure called a Stupa as a shrine, let's consider the provided context and the information we need to extract:\n\n- Stupa is described as a \"mound-like or hemispherical structure containing relics (such as śarīra – typically the remains of Buddhist monks or nuns) that is used as a place of meditation.\"\n- The context mentions that \"Religious buildings in the form of the Buddhist stupa, a dome shaped monument, started to be used in India as commemorative monuments associated with storing sacred relics of the Buddha.\"\n- Additionally, \"Shanti Stupa is a Buddhist white-domed stupa (chorten) on a hilltop in Chanspa, Leh district, Ladakh, in the north Indian state of Jammu and Kashmir.\"\n\nFrom these points, we can clearly see that the Stupa is a religious structure used by Buddhism, specifically for storing relics and as a shrine.\n\n\n{reasoning process} We are asked which religion uses a domed structure called a Stupa as a shrine. From the given context, it is clear that the Stupa is associated with Buddhism and is used as a shrine. {####} Buddhism ✅

Figure 18: Correct reasoning example 2 in HRPO.

**Query**: John fills a 6 foot by 4 foot pool that is 5 feet deep. It cost \$.1 per cubic foot to fill. How much does it cost to fill?

**HRPO (Tokens in Hybrid Rollout):** To find out how much it costs to fill the pool, we need to first calculate the volume of the pool. The volume of a rectangular prism (like a pool) is given by the formula:%n%nVolume = Length × Width × Height%n%nIn this case, the length is 6 feet, the width is 4 feet, and the height is 5 feet. So,%n%nVolume = 6 × 4 × 5 = 120 cubic feet%n%nThe cost to fill the pool is given as $0.1 per cubic foot. Therefore,%n%nCost = Volume × Price per cubic foot%nCost = 120 × $0.1 = $12%n%nSo, the cost to fill the pool is $12. {reasoning process} #### {12} ✅

Figure 19: Correct reasoning example 3 in HRPO.

**Query**: The sum of the proper divisors of 18 is 21. What is the sum of the proper divisors of 198?

**HRPO (Tokens in Hybrid Rollout):** To find the sum of the proper divisors of 198, we first need to identify all the proper divisors of 198. Proper divisors are the divisors of a number excluding the number itself.\n\n198的因数有：1, 2, 3, 6, 9, 11, 18, 22, 33, 66, 99, 198\n除去198本身，其余的数都是198的因数。\n\n所以，198的proper divisors是：2, 3, 6, 9, 11, 18, 22, 33, 66, 99。\n\n接下来，我们计算这些数的和：\n1 + 2 + 3 + 6 + 9 + 11 + 18 + 22 + 33 + 66 + 99 = 270\n\n因此，198的proper divisors的和是270。{reasoning process} #### {answer} 270 ✅

Figure 20: Correct reasoning example 4 in HRPO.

**Query**: What is $(7^{-1})^{-1}$?

**HRPO (Tokens in Hybrid Rollout):** To find the value of %%((7ˆ{-1})ˆ{-1})%%), we need to apply the rules of exponents. (Note: The exponent rule %%((-a)ˆ{-1} = -aˆ{-1}%%) but )%nI am_____ (leave the word in the blank) and the study of music will be called ____. {reasons} to____. For this, the Rhythm Method is the only way to teach music.%n{reasoning process} #### To understand the reasoning process behind the given problem, let's break down the components and mathematical operations involved.%n%nFirst, we are dealing with %%((7ˆ{-1})ˆ{-1}%%). %n%nBy exponent rules, %%((aˆ{-m})ˆ{-n} = aˆ{-m %Íot -n} = aˆ{mn}%%). Therefore, %%((7ˆ{-1})ˆ{-1} = 7ˆ{(-1) %Íot -1} = 7ˆ1 = 7%%).%n%nNext, we are told that the Rhythm Method is the only way to teach music. However, the blank in the Rhythm Method is left incomplete with a word, which leaves the option to use any word that fits the context, such as 'method', 'system', 'approach', 'tool', 'technique', or 'strategy'.%n%nGiven these points, we can conclude:%n%n{reasoning process} #### %%(7%%) ✅

Figure 21: Correct reasoning example 5 in HRPO.

**Query**: What party was Johan Henrik Deuntzer a part of before the party that translates to the "Radical Left?"?

**HRPO (Tokens in Hybrid Rollout):** Social Democratic League #### Social Democratic League {reasoning process}：据问，Johan Henrik Deuntzer before the party that translates to the \"Radical Left.\" 但是，根据提供的信息，没有明确提到Johan Henrik Deuntzer在之前是否属于Social Democratic League。因此，我们无法确定答案。{Johan Henrik Deuntzer} #### Johan Henrik Deuntzer ❌

Figure 22: Mistaken reasoning example 1 in HRPO.

**Query**: When the Marlins are the home team, games are played at this park which was LEED certified as the greenest MLB park in what year?

**HRPO (Tokens in Hybrid Rollout):** 2012 {reasoning process} 首先，根据上下文，我们需要找到 Marlins Park 在什么年份成为 LEED 金牌认证的绿州球场。然后，我们需要找到 Marlins Park 是什么时候成为 MLB 球队的绿州球场的。这两个信息在给定的上下文中有提到。最后，我们需要将这两个关键信息联系起来以创建一个完整的问题。{answer} 2012 年，Marlins Park 成为 MLB 球队中成为 LEED 金牌认证的绿洲球场的年份。 ✗

Figure 23: Mistaken reasoning example 2 in HRPO.

**Query**: Can you name the singer of the title track for '1971 - Diamonds Are Forever'?

**HRPO (Tokens in Hybrid Rollout):** Thought: 根据提供的背景信息,我们得知\"1971 - Diamonds Are Forever\"是詹姆斯·邦德第七部电影的主题曲。让我们逐一分析各个电影的主题曲:\n\n1. 第一部 - \"Goldfinger\"由艾玛·罗伯茨演唱\n2. 第二部 - \"Thunderball\"由罗伯特·施莱格尔演唱\n3. 第三部 - \"For Your Eyes Only\"由约翰·哈里特演唱\n4. 第四部 - \"The Spy Who Loved Me\"由约翰·赫特演唱\n5. 第五部 - \"Moonraker\"由布莱恩·特里菲特演唱\n6. 第六部 - \"On Her Majesty's Secret Service\"由安格丽卡·休斯顿演唱\n7. 第七部 - \"Diamonds Are Forever\"由西尔维娅·布拉西演唱\n\n因此,答案是西尔维娅·布拉西(Silvia Braithwaite) #### Silvia Braithwaite\n\nWestlife is a popular Irish rock band from Northern Ireland, so it's possible that one of the singers from Westlife could have performed the song as well. However, the question asks for the singer of the title track for the 1971 film \"Diamonds Are Forever\", and the answer is the one mentioned in the background information. Therefore, the singer of the title track for '1971 - Diamonds Are Forever' is Shirley Bassey. ✗

Figure 24: Mistaken reasoning example 3 in HRPO.

**Query**: What is the largest prime factor of 1391?

**HRPO (Tokens in Hybrid Rollout):** To find the largest rank for the first emperor, placing parentheses. The largest rank of the integral is 1391. Let's first factorize 1391 into its prime factors.%n%n1391 is an odd number, so it is not divisible by 2. The sum of the digits of 1391 is (1 + 3 + 9 + 1) = 14, which is not divisible by 3, so 1391 is not divisible by 3. It is not divisible by 5 either, but it is divisible by 7: 1391 ÷ 7 = 198.3571. The product of the digits of 1391 is 1 × 3 × 9 × 1 = 27, which is divisible by 3, so 1391 is divisible by 3. Therefore, 1391 = 3 × 463. %n%nNow we have the prime factorization of 1391 as 1391 = 3 × 463. Next, we need to determine if either 3 or 463 is a prime number. The numbers 3 and 463 are both integers. A series of operations involving integers. Therefore 3 is divisible by 3, so it is not prime._____. 3 and 463 are not prime, but the number 463 is a prime number, where 463 is a prime number since it has no positive integer divisors other than 1 and itself.能量大能量大能量大能量大能量大能量大能量大能量大能量大能量大能量大能量大能量大能量大能量大能量大能量大能量大能量大能量大能量大能量大能量大能量大能量大能量大能量大能量大能量大能量大能量大能量大…… ✗

Figure 25: Mistaken reasoning example 4 in HRPO.

**Query**: There exist nonzero integers $a$ and $b$ such that the quadratic\n\\[(ax - b)^2 + (bx - a)^2 = x\\]has one integer root and one non-integer root. Find the non-integer root.

**HRPO (Tokens in Hybrid Rollout):** To find the non-integer root of the equation \\((ax - b)^2 + (bx - a)^2 = x\\), let's first simplify and analyze the equation. \n\nLet \\( y_1 = ax - b \\) and \\( y_2 = bx - a \\). The given equation transforms into \\( y_1^2 + y_2^2 = x \\).\n\nFrom here, we need to explore the nature of \\(x\\), which is a function of \\(y_1\\) and \\(y_2\\). \n\nGiven the equation \\( y_1^2 + y_2^2 = x \\), we know that \\(x\\) is a function of the combination of \\(y_1\\) and \\(y_2\\), and since \\( y_1 \\) and \\( y_2 \\) are linear in terms of \\(a\\) and \\(b\\), we need to analyze the roots.\n\nTo find the roots, we need to find the values of \\(x\\) that make the quadratic equation true. Let's solve for \\(a\\) and \\(b\\) in terms of \\(x\\).\n\nIf we assume \\( (ax - b)^2 + (bx - a)^2 = x \\), let's assume a value for \\(x\\) and check if the roots fit the criteria. \n\nFor simplicity, let's check if there's a value of \\(x\\) such that the roots are one integer and one non-integer. \n\nOne way to approach this is to notice that the sum of squares \\(y_1^2 + y_2^2 = x \\) will be satisfied if \\(x \\) itself is a sum of two squares of integers. By trying out some values, we can find out the roots.\n\nLet's assume the non-integer root, we denote it by \\((n\\)), and solve for it. \n\nFor example, if we assume \\(a = 1\\) and \\(b = 2\\), then we get:\n\\((x - 2)^2 + (2x - 1)^2 = x\\).\n\nExpanding this, we get \\(x^2 - 4x + 4 + 4x^2 - 4x + 1 = x\\), which simplifies to \\(5x^2 - 8x + 5 = x\\) or \\(5x^2 - 9x + 5 = 0\\).\n\nThe roots of this quadratic equation can be found using the quadratic formula \\(x = \\frac{-b \\pm \\sqrt{b^2 - 4ac}}{2a}\\).\n\nHere, the non-integer root \\((n\\)) is the root closest to but not equal to an integer.\n\nSo, the non-integer root \\((n\\)) in the equation, \\(n = \\frac{9 \\pm \\sqrt{81 - 100}}{10}\\), which simplifies to \\(n = \\frac{9 \\pm \\sqrt{-19}}{10}\\).\n\nHowever, \\( \\sqrt{-19}\\) is not real, so we conclude that there must exist a non-integer root \\((n\\)) that satisfies the given equation.\n\nNow, let's calculate the value of \\((n\\)) in the equation, \\((n\\)) = \\((\\frac{9 \\pm \\sqrt{81 - 100}}{10}\\), which simplifies to \\(n = \\frac{9 \\pm \\sqrt{-19}}{10}\\).\n\nSince中, 考虑到 \\(a\\) 和 \\(b\\) 的值, 设我们观察到非 \\((n\\)) = \\((\\frac{9 \\pm \\sqrt{81 - 100}}{10}\\), 简化到 \\(n = \\frac{9 \\pm \\sqrt{-19}}{10}\\)。\n\n但是, \\(\\sqrt{-19}\\) 不是实数, 所以, 实际上不存在这样的 \\((n\\))。\n\n因此, 我们观察到, 该非-整数根 \\((n\\))满足题目的条件。{因此, 题目本身即不存在满足条件的非整数根\\((n\\)), 即满足题目的条件。\n\n{reasonably,我们可以观察到的非整数根 \\((n\\))满足题目的条件, 即 \\(n = \\frac{9 \\pm \\sqrt{81 - 100}}{10}\\)。\n\n根据题目条件, 我们考虑一个满足条件的非整数根 \\((n\\)) 。\n\n{这样, 非整数根 \\((n\\)) 要求一个非整数根 \\((n\\)) 满足题目的条件。即 \\(n = \\frac{9 \\pm \\sqrt{81 - 100}}{10}\\)。\n\n考虑到 \\((n\\)) 满足题目的条件, 我们观察到的非整数根 \\((n\\)) 满足题目的条件, 即 \\(n = \\frac{9 \\\n

Figure 26: Mistaken reasoning example 5 in HRPO.

