# OpenReview forum: "Hybrid Latent Reasoning via Reinforcement Learning"
_NeurIPS.cc/2025/Conference — NeurIPS 2025 poster_

### Official Review · Reviewer_f57F · 2025-06-28

**Clarity:** 2
**Significance:** 2
**Originality:** 3
**Rating:** 4
**Confidence:** 4

**Summary:**

This paper introduces HRPO using RL to train LLMs for hybrid latent reasoning. The hidden state $h_t$ and token embedding $ê_{t+1}$ exist in different vector spaces and cannot be directly combined. The paper proposes projecting $h_t$ back into the token embedding space by using the predicted next-token probability distribution $p_{t+1}$ to compute a weighted average of the vocabulary embeddings $W_e$. After obtaining compatible inputs, HRPO dynamically blends them via a gating mechanism to produce the final next-step input $e_{t+1}$. Entirely RL-based (i.e., GRPO), HRPO requires no CoT annotations and learns through RL with simple outcome rewards. The latent space encodes information to enable HRPO to generate more concise reasoning steps than traditional CoT.

**Questions:**

1. What are training time and computational resource requirements?
2. The model performs best when relying heavily on either tokens or latent states but struggles in intermediate regimes. What explains this bimodal behavior? Could it reflect fundamentally distinct reasoning modes for different tasks?
3. Can HRPO generalize to unseen tasks without RL fine-tuning? How might it scale to larger models (70B+)? Would the optimal gating balance shift with model size?
5. Has the projected hidden state  been analyzed? For math tasks, do these vectors cluster near relevant numbers/operators in embedding space? Visualizations or clustering analyses could clarify latent space properties.

**Ethical Concerns:**

["NO or VERY MINOR ethics concerns only"]

**Final Justification:**

I thank the authors for providing the response. The rebuttal addresses most of the key concerns I initially raised. Based on the authors' response, I increase my rating for the paper.

The authors clarify the distinction between  `surface interpretability`  and `intrinsic abstraction`  This resolves the logical inconsistency, while also clearly defining the boundaries of the method's interpretability – the true reasoning process occurs within the abstract latent space. This point is now clear, and I consider this a reasonable trade-off.  The authors also commit to including a t-SNE/PCA visualization analysis of the latent space in the final version. And I suggest that the authors include the computational overhead in their paper.

**Limitations:**

yes. But I did not find explicit limitations or broader impact section.

**Paper Formatting Concerns:**

Results comparing the performance of HRPO with other alternative latent reasoning methods should be placed in the main paper.

**Quality:**

3

**Strengths And Weaknesses:**

Strengths:
1. Combines RL, gating mechanisms, and latent reasoning while addressing key limitations of existing latent reasoning methods (e.g., reliance on CoT data, incompatibility with LLMs).
2. Clear structure, with intuitive visualizations (Figures 2 and 3) highlighting the method core ideas and advantages.
3. Extensive experiments across benchmarks and ablation studies demonstrate superior performance.

Weaknesses:
1. For large models, their native autoregressive reasoning might already suffice. It maybe making HRPO marginal benefits negligible.
2. HRPO primarily combines existing techniques: latent reasoning, gating mechanisms (from GRU/LSTM), and KL-penalized RL policy gradients (from GRPO).
3. HRPO claims interpretability, but cross-lingual examples suggest abstract, language-independent internal representations. How does this `surface interpretability` (token traces) relate to `intrinsic abstraction` (latent reasoning)? Does the actual reasoning occur in an opaque latent space?

---

> ### Author Rebuttal · Authors · 2025-07-31
>
> We thank you for the insightful feedback and are excited that you find our work “addressing key limitations of existing latent reasoning” and “demonstrate superior performance”. Below, we respond to your questions and weaknesses, we hope these could address your concerns regarding our paper:
>
> **W1. For large models, their native autoregressive reasoning might already suffice...**
>
> - While post-trained LLMs do possess strong native autoregressive reasoning, HRPO still yields consistent & substantial gains by unlocking their hybrid representations. For example, compared with larger LLMs or further RL baselines, HRPO (with the 3B backbone) achieves the highest average EM of 0.378, outperforming a much larger Qwen2.5‑7B baseline at 0.335. Similar performance gains can be found in our STEM experiments, where HRPO demonstrates both improved performance and intriguing reasoning patterns. These results demonstrate that hybrid latent reasoning remains beneficial even when autoregressive reasoning is already strong, providing novel directions for future research in reinforcement learning and LLM post-training.
>
> **W2. HRPO primarily combines existing techniques...**
>
> - We appreciate the reviewer’s feedback on HRPO’s design and respectfully disagree with it. We argue that although HRPO builds on existing foundations in reinforcement learning, recurrence modeling and latent reasoning, we propose a first‑of‑its‑kind generation / training paradigm by learning an optimal combination of continuous and discrete representations. The proposed HRPO is not only novel, but also carefully designed for optimal performance and efficiency. In particular, we overcome several key limitations in our HRPO implementation: (1) projection step: maps hidden states back into the token embedding manifold in a differentiable way while preserving the model’s generative capabilities. (2) learnable gating (Eq. 4): anneals from predominantly token embeddings to richer hybrid inputs, enhancing the training dynamics and utilization of reward signals. (3) on‑policy learning: simple RL objective that fuses discrete and continuous trajectories, enabling accurate policy gradients without a separate value function. The novel hybrid design along with optimal implementation choices of HRPO yields a robust, resource‑efficient framework that consistently outperforms both pure autoregressive and latent‑only approaches across diverse benchmarks, demonstrating HRPO as a scalable, high‑performance paradigm for RL‑based hybrid post‑training.
>
> **W3. HRPO claims interpretability, but cross-lingual examples suggest abstract…**
>
> - We thank you for the insightful comment. Indeed, HRPO remains surface‑interpretable through its token traces while harnessing abstract latent reasoning. The sampled tokens (and probability distributions of latent tokens) still form coherent chains‑of‑thought, making the model’s reasoning path inspectable. And the latent representations are learnt with the objective of maximizing the outcome rewards rather than explainability, which may account for the cross‑lingual patterns that enable token mixing across languages. Conversely, RL‑based latent reasoning approaches may not yield true intrinsic abstractions, since they optimize solely for reward maximization and high‑probability token outputs without explicitly learning interpretable hybrid representations. Consequently, HRPO prioritizes reasoning efficacy (similar to recent efforts in RLVR) by employing latent features to enrich generation performance, with interpretability naturally emerging from the sampled token‑level outputs.
>
> **Q1. What are training time and computational resource requirements?**
>
> - We appreciate the question and agree that training efficiency is critical for real‑world deployment of HRPO. As such, we would like to highlight that: (i) our gating mechanism adopts the straightforward recurrent‑modeling design [1-2], which adds minimal compute cost at inference (limited to decoding steps, while prefill processes just the prompt tokens). Thus in our experiments, we find no noticeable effect on inference speed across model sizes or generation lengths (see details in our response to Q1); (ii) compared to existing RL algorithms, HRPO only introduces negligible parameters in its gating (in millions compared to billions of LLM sizes), and thus the resulting model sizes, training speed as well as VRAM footprint remain virtually identical from GRPO; (iii) we have implemented parameter efficient / full fine‑tuning with optimized kernels, where the models are patched for efficient rollout and policy update, we also find that for post-training with 8‑bit AdamW could further reduce both memory and compute overhead. Therefore in our experiments, we observe negligible difference of training and inference compute due to the minimal design of HRPO. As an illustration, we evaluate GRPO and HRPO on the 1.5B backbone using the GSM8k and MATH benchmarks, reporting training throughput, peak VRAM usage, and inference latency on their evaluation sets. Despite adding the gating mechanism, HRPO incurs only a minor compute and memory overhead while matching GRPO’s inference speed. These differences are negligible in practice, suggesting HRPO's efficiency and its potential to be trained using only consumer hardware.
>
> - | Metric                          | GRPO    | HRPO    |
> |---------------------------------|---------|---------|
> | GSM8k Training Throughput       | 269.88  | 262.21  |
> | GSM8k Training VRAM (max)       | 11.082  | 11.536  |
> | GSM8k Inference Time            | 16:25   | 16:21   |
> | MATH Training Throughput        | 228.97  | 225.86  |
> | MATH Training VRAM (max)        | 14.154  | 15.222  |
> | MATH500 Inference Time          | 14:15   | 14:04   |
>
> **Q2. The model performs best when relying heavily on either tokens or...**
>
> - Our results in Table 3 reveal a bimodal pattern: performance surges at low and high hidden‑ratio settings but dips around the midpoint. We interpret this as two complementary reasoning regimes: token‑dominant mode generates detailed, step‑by‑step CoT, ideal for tasks demanding precise, literal reasoning. On the other hand, latent‑dominant mode leverages compact, context‑summarizing hidden representations, which excel on abstract or knowledge‑intensive problems (e.g., RAG). Balancing both modes equally appears to dilute each regime’s strengths, introducing interference rather than synergy. We also note that our exploration of these hybrid representations is still in its early-stage, as these are learned solely through reward maximization, and a deeper analysis of their behavior and interpretability remains an important direction for future work.
>
> **Q3. Can HRPO generalize to unseen tasks without RL fine-tuning?...**
>
> - Concurrent to our paper, [1-2] show that latent representations could generalize to unseen tasks for large‑scale post-trained models even without fine-tuning. While these findings establish zero‑shot latent reasoning as viable for post‑trained LLMs, RL‑based fine‑tuning remains impractical as it requires stochastic sampling to optimize the policy model. Therefore, the proposed HRPO represents the best of both worlds by introducing hybrid representations with stochastic token sampling. In addition, HRPO focuses on the algorithmic design of RL-based latent reasoning, and thus remains domain‑agnostic and seamlessly supports generalization via reinforcement learning. For example, we train a single model on the combined MMLU auxiliary dataset and ARC training set, then evaluate it on both benchmarks in our experiments. Similarly, we post-train one model solely on the NQ + HotpotQA data mixture and assess it on multiple knowledge-intensive tasks. These results demonstrate that HRPO can be easily extended to multi‑task / domain generalization settings. Paired with our careful design and implementation, RL-based hybrid reasoning delivers both superior performance and enhanced efficiency on reasoning‑intensive tasks. Regarding larger models, we are unable to perform extensive RL training on large models across all selected datasets due to our limited time and compute resources, we leave this exploration and the analysis of the optimal gating balances across model scales as future work of HRPO.
>
> **Q4. Has the projected hidden state been analyzed?...**
>
> - While we did not perform explicit clustering of the projected embeddings in this work, our qualitative examples (as in Sec. 4 and App. C) reveal that these vectors encode semantically meaningful information: they surface as language‑agnostic token traces and shorten completion lengths when rich context is captured. We also agree that visualized analysis could provide valuable insights of HRPO, especially for numeric and operator tokens in math tasks. Accordingly, we will incorporate additional visualizations (e.g., t‑SNE or PCA of the learnt latent representations) and in‑depth discussions of their structural and semantic properties. We hope these clarifications and analyses address the reviewer’s concerns and highlight HRPO’s contributions.
>
> [1] Zhang, Zhen, et al. "Soft thinking: Unlocking the reasoning potential of llms in continuous concept space." arXiv preprint arXiv:2505.15778 (2025).
>
> [2] Zhuang, Yufan, et al. "Text generation beyond discrete token sampling." arXiv preprint arXiv:2505.14827 (2025).

---

### Official Review · Reviewer_K5az · 2025-06-30

**Clarity:** 3
**Significance:** 2
**Originality:** 3
**Rating:** 3
**Confidence:** 3

**Summary:**

The paper introduces Hybrid Reasoning Policy Optimization (HRPO), a novel reinforcement learning (RL)-based framework designed to enhance latent reasoning in large language models (LLMs). Unlike traditional autoregressive chain-of-thought (CoT) methods, HRPO integrates continuous hidden state representations with discrete token sampling, leveraging a gating mechanism to balance these representations. The approach aims to utilize LLMs' intrinsic reasoning capabilities without relying on CoT trajectories, thereby reducing training costs and improving adaptability. The authors claim HRPO outperforms existing methods on knowledge- and reasoning-intensive tasks, maintains interpretability, and exhibits unique behaviors such as cross-lingual patterns and shorter completion lengths.

**Questions:**

1) Given that computational efficiency is a critical concern for reasoning in large language models, why was there no comparative analysis of HRPO's efficiency against autoregressive CoT methods? Could the authors provide data on training time, inference latency, or computational resource requirements?
2)  How is the gating mechanism optimized during RL training? Are there specific hyperparameters or loss functions used to control the balance between discrete and continuous representations?

**Ethical Concerns:**

["NO or VERY MINOR ethics concerns only"]

**Limitations:**

See the Questions.

**Quality:**

2

**Strengths And Weaknesses:**

Strengths
1) HRPO is the first approach to apply reinforcement learning to hybrid latent reasoning, combining discrete token sampling with continuous hidden state representations. This novel framework leverages stochasticity from token sampling to enable RL-based optimization, bypassing the need for CoT trajectories.
2) By utilizing LLMs' inherent reasoning patterns, HRPO reduces reliance on extensive CoT annotations, which are computationally expensive and often unavailable. This makes the approach more practical for broader applications.
3) HRPO-trained models remain interpretable and exhibit intriguing behaviors, such as cross-lingual reasoning patterns and shorter completion lengths, suggesting practical benefits beyond performance metrics.


Weaknesses
1) The paper does not provide a comparative analysis of computational efficiency, a critical factor for practical deployment. The hybrid approach, which combines explicit token sampling with latent reasoning via a gating mechanism, may incur higher computational costs compared to traditional CoT methods, yet this is not addressed.
2) While the gating mechanism is a core component, the paper lacks details on how it is optimized during training. The process of learning to incorporate richer hidden state features is vaguely described, making it difficult to assess its robustness.
3) The paper claims HRPO enhances adaptability but provides limited discussion on specific real-world applications where its hybrid reasoning approach offers clear advantages over existing methods, reducing its perceived practical value.

---

> ### Author Rebuttal · Authors · 2025-07-31
>
> Thank you for your constructive review, we’re delighted you find our work “novel”, “practical”, “exhibit intriguing behaviors” and “outperforms existing methods”. In the following, we present detailed responses and additional experimental results addressing the weaknesses you raised, and we hope these clarifications could resolve your concerns:
>
> **W1. The paper does not provide a comparative analysis of computational efficiency...**
>
> - While our primary focus was on demonstrating the effectiveness of HRPO in improving reasoning performance and reasoning efficiency, we agree that training efficiency is critical for real‑world deployment. As such, we would like to highlight that: (i) our gating mechanism adopts the straightforward recurrent‑modeling design [1-2], which adds minimal compute cost at inference (limited to decoding steps, while prefill processes just the prompt tokens). Thus in our experiments, we find no noticeable effect on inference speed across model sizes or generation lengths (see details in our response to Q1); (ii) compared to existing RL algorithms, HRPO only introduces negligible parameters in its gating (in millions compared to billions of LLM sizes), and thus the resulting model sizes, training speed as well as VRAM footprint remain virtually identical from GRPO; (iii) we have implemented parameter efficient / full fine‑tuning with optimized kernels, where the models are patched for efficient rollout and policy update, we also find that for post-training with 8‑bit AdamW could further reduce both memory and compute overhead. We hope these points clarify HRPO’s efficiency and its negligible compute overhead relative to standard RL-based reasoning.
>
> **W2. While the gating mechanism is a core component...**
>
> - We thank you for raising this point and explain the training process of HRPO in the following. Concretely, the gating and the LLM parameters are learned “end‑to‑end” under our RL objective (Eq. 6), which resembles the REINFORCE algorithm augmented with grouped rewards and standardized advantage estimation [3-4]. In each training iteration, the LLM first generates multiple rollouts for each of the input queries, where the rollouts are evaluated with the reward function to compute their respective advantages. Next, we estimate the policy gradients of HRPO as per Eq. 6, which are used to update the LLM and gating parameters with the optimizer (e.g., AdamW). Aside from the additional gate and our novel hybrid generation strategy, HRPO’s training loop follows the same model‑based RL workflow (e.g., PPO, GRPO) common in RLHF / RLVR. We will expand the descriptions of our training process with more details in our revision, and we hope this could clarify your concerns regarding the training of HRPO.
>
> **W3. The paper claims HRPO enhances adaptability but provides limited discussion on specific real-world applications...**
>
> - We appreciate your valuable comment and expand on practical scenarios where HRPO’s hybrid latent reasoning is uniquely beneficial. We believe HRPO could be particularly useful in the following scenarios: (i) low‑annotation domains: by eliminating reliance on human‑annotated CoT traces, HRPO is ideal for training LLMs in specialized domains where step‑by‑step rationales are scarce or costly. (ii) cost‑sensitive training / inference: the efficient training framework and shorter completion lengths reduce both training costs and inference latency in production settings. (iii) strong reasoning required: on STEM problems and complex multi-hop queries, HRPO’s latent representations encode intermediate reasoning effectively and remain interpretable, potentially improving user’s step‑by‑step feedback efficiency. We thank you for the suggestion and will include additional discussions on the real-world applications of HRPO in the final manuscript to emphasize these points.
>
> **Q1. Given that computational efficiency is a critical concern…**
>
> - In our experiments, we observe negligible difference of training and inference compute due to the minimal design of HRPO. As an illustration, we evaluate GRPO and HRPO on the 1.5B backbone (with 8bit AdamW & LoRA, more details are reported in the experiment settings in our technical appendix) using the GSM8k and MATH benchmarks, reporting training throughput, peak VRAM usage, and inference latency on their evaluation sets. Despite adding the gating mechanism, HRPO incurs only a minor compute and memory overhead while matching GRPO’s inference speed. These differences are negligible in practice, suggesting that HRPO achieves its reasoning improvements with comparable training and inference costs.
>
> - | Metric                          | GRPO    | HRPO    |
> |---------------------------------|---------|---------|
> | GSM8k Training Throughput       | 269.88  | 262.21  |
> | GSM8k Training VRAM (max)       | 11.082  | 11.536  |
> | GSM8k Inference Time            | 16:25   | 16:21   |
> | MATH Training Throughput        | 228.97  | 225.86  |
> | MATH Training VRAM (max)        | 14.154  | 15.222  |
> | MATH500 Inference Time          | 14:15   | 14:04   |
>
> **Q2. How is the gating mechanism optimized during RL training?…**
>
> - We thank you for the question. In HRPO, the gating parameters are randomly initialized and start by favoring the token embeddings to preserve the generative capabilities. In training, the gating weights are learned “end‑to‑end” alongside the LLM parameters via the same policy gradient objective used for the rest of HRPO as in Eq. 6. In other words, we generate rollouts to select high‑reward responses, then minimize their negative log‑probabilities (weighted by advantages) to compute policy gradients. These gradients update both the LLM and gating parameters, with the objective of finding an optimal policy that produces semantically rich hybrid representations for reasoning traces. As for the hyperparameter, we tune $r_{min}$ to control the balance between discrete and latent features (i.e., the lower bound of the gating component’s sampling range). With higher ​$r_{min}$, this ensures that the model draws predominantly from token embeddings, whereas lower $r_{min}$ encourages exploration in the hybrid space and the learning of informative latent representations. We also provide a detailed explanation of our training pipeline in our response to weakness 2 and hope these could clarify the learning of our gating mechanism, we remain available for discussion in case of any further concerns / questions.
>
>
> [1] Orvieto, Antonio, et al. "Resurrecting recurrent neural networks for long sequences." International Conference on Machine Learning. PMLR, 2023.
>
> [2] De, Soham, et al. "Griffin: Mixing gated linear recurrences with local attention for efficient language models." arXiv preprint arXiv:2402.19427 (2024).
>
> [3] Sutton, Richard S., et al. "Policy gradient methods for reinforcement learning with function approximation." Advances in neural information processing systems 12 (1999).
>
> [4] Shao, Zhihong, et al. "Deepseekmath: Pushing the limits of mathematical reasoning in open language models." arXiv preprint arXiv:2402.03300 (2024).

---

> > ### Comment · Reviewer_K5az · 2025-08-08
> >
> > I'd like to thank the authors for the response which have addressed most of my concerns. I have accordingly adjusted the score.

---

> > > ### Author Response · Authors · 2025-08-08
> > > **Response to Reviewer**
> > >
> > > We appreciate your insightful feedback and your acknowledgment that our response addressed most of your concerns, and are grateful for the time and effort you have dedicated to reviewing our work. We also kindly remind you to adjust the score accordingly, as mentioned in your comment, thank you in advance!

---

> ### Author Response · Authors · 2025-08-06
> **Discussion Reminder**
>
> Dear reviewer, we thank you for the time and efforts you have dedicated to reviewing our work! As the discussion period is ending today, we kindly remind you to acknowledge our response to your reviews. We hope our previous responses could address your concerns and are happy to follow-up in further discussion if any concerns remain. Thank you!

---

> ### Author Response · Authors · 2025-08-08
> **Discussion Reminder (cont.)**
>
> We sincerely appreciate the time and effort you have dedicated to reviewing our work. As the discussion period is ending, we kindly remind you to acknowledge our rebuttal. We hope our earlier responses have addressed your concerns, and we are happy to engage in further discussion if any questions remain. Thank you!

---

> ### Author Response · Authors · 2025-08-09
> **Response to Reviewer (cont.)**
>
> With only a few hours remaining in the reviewer–author discussion period, we sincerely remind you to adjust your score accordingly, as noted in your earlier comment. Thank you again for your thoughtful feedback and for acknowledging our rebuttal!

---

### Official Review · Reviewer_tnmr · 2025-07-02

**Clarity:** 3
**Significance:** 3
**Originality:** 3
**Rating:** 5
**Confidence:** 3

**Summary:**

To tackle known limitations of latent reasoning in LM training (e.g., incompatibility of output and input embeddings), the authors propose a gating mechanisms that gradually incorporates hidden states from previous steps into sampled tokens that is trained with an RL algorithm (“hybrid reasoning policy optimization”, HRPO), which is derived from GRPO by omitting likelihood ratios, due to the rollout nature of having token and latent representation rollouts. The derivation of HRPO and the underlying gating mechanism uses a sampling free mapping to allow for differentiability and a learnable gating parameter vector.

The performance of HRPO is evaluated on knowledge benchmarks (NQ, TriviaQA, HotpotQA, 2WikiMQA, and Bamboogle) and STEM benchmarks (GSM8k, MATH, MATH500, MMLU-STEM, ARC-Challenge). As baselines, the authors compare to more capable models (7B parameters) with RAG and COT as well as SFT, RAG, PPO, and GRPO baselines for the smaller models also trained with HRPO (Qwen2.5 instruct 1.5B and 3B).

Without reporting statistical significance analysis or uncertainty estimates, the authors present results that indicate a strong performance of HRPO compared to the baselines. In addition, the authors study the impact of HRPO gating parameter initialization and sensitivity and qualitative reasoning studies.

The submission does not explicitly discuss limitations or has an impact statement.

**Questions:**

- In Figure 3, why do the GRPO and HRPO plots share almost all artifacts (they look like they are perfectly in synch for most jumps in either direction).
- Is there an experiment that could test the hypothesis in lines 268-270?
- Have you trained the same model for both tasks with HRPO? Showing that you can train HRPO for multipel tasks simultaneously would demonstrate that it can work without the gating mechanisms, e.g., introducing negative interference.
- What are the limitations of your submission?
- What are the potential societal impacts of HRPO? For example, can HRPO and latent reasoning training make evaluations of models for safety-critical applications harder as the models learnt to be less verbose about their reasonings (just an arbitrary example)?

**Ethical Concerns:**

["NO or VERY MINOR ethics concerns only"]

**Final Justification:**

The authors were able to address my concerns and I believe the proposed HRPO has the potential to influence and guide future work in this field.

**Limitations:**

No (see questions for feedback).

**Paper Formatting Concerns:**

No.

**Quality:**

3

**Strengths And Weaknesses:**

Reasons to accept:

- The paper is well written and, to the best of my knowledge, original work (Clarity).

- The authors provide good theoretical description and background for the introduction of HRPO and conduct a series of experiments to make an argument for all their claims (Quality).

- The idea of using a gating mechanism is elegant and promising. If all claims are true, the submission has the potential to offer a new way of training models on their latent reasoning in an RL setting (Significance).

Reasons to reject:

- The authors do not conduct any statistical significance analysis whatsoever and do not report statistical uncertainties of their results. Their results are promising and I would like to award a higher score, but due to good scientific practice, it is unclear whether the reported improvements over the baselines are statistically significant and whether the discussed claims in section 4.1 and 4.2 are valid. This concern can be fully addressed by reporting, e.g., 95% confidence intervals for all results in Table 1, Table 2, and either conducting time-series uncertainty estimates (e.g., Gaussian processes) or clearly stating that the results in the Figures 3, 4, and 5 are only statistical trends, not results. If the improvements in Table 1 and 2 are statistically significant, I will happily increase my score.
- The authors train the gating mechanism/HRPO separately for each set of tasks, makign it ambiguous whether you can use it to train models across tasks.
- The lack of a limitation section with a balanced discussion as well as the lack of an impact statement weakens the position of the submission.

---

> ### Author Rebuttal · Authors · 2025-07-31
>
> We appreciate the insightful and constructive review and are thrilled that our work is regarded as “elegant”, “promising”, “original” and “well written”. Below, we provide our responses along with additional results to respond the mentioned weaknesses and questions:
>
> **W1. The authors do not conduct any statistical significance analysis.…**
>
> - We acknowledge your concerns w.r.t. the statistical significance of our HRPO results in comparison to baseline methods. This stems from (i) our focus on a simple hybrid RL implementation using REINFORCE with group-relative rewards [1–2], which does not include optimization techniques like per-token rewards or ratio clipping; and (ii) we follow the standard practice of using greedy decoding for pass@1 evaluation [3–5], ensuring our results are easy to evaluate and reproducible. To address your concerns, we conduct additional sampling-based evaluations on the STEM tasks, which are often more challenging and exhibit greater variance. We present the averaged results below, with statistical significance (paired t-test) highlighted in bold:
>
> - | 1.5B | GSM8k | MATH  | MATH500 | MMLU-ST | ARC-C |
> | ------ | ----- | ----- | ------- | ------- | ----- |
> | PPO    | 0.701 | 0.505 | 0.511   | 0.551   | 0.716 |
> | GRPO   | 0.710 | 0.510 | 0.512   | 0.554   | 0.722 |
> | HRPO   | 0.712 | **0.515** | 0.517   | **0.565**   | **0.731** |
>
> - | 3B | GSM8k | MATH  | MATH500 | MMLU-ST | ARC-C |
> | ------ | ----- | ----- | ------- | ------- | ----- |
> | PPO    | 0.825 | 0.597 | 0.600   | 0.574   | 0.802 |
> | GRPO   | 0.827 | 0.595 | 0.599   | 0.577   | 0.808 |
> | HRPO   | **0.838** | **0.606** | **0.609**   | **0.585**   | 0.815 |
>
> - Across both backbones, HRPO outperforms PPO and GRPO on all benchmark datasets. For the 1.5B backbone HRPO, paired t-tests show these gains reach statistical significance (p < 0.05) in three out of five tasks. With the 3B model the improvements are even larger (+1.4%) respectively, and four out of five comparisons are statistically significant. These results demonstrate that our simple hybrid-RL framework HRPO not only delivers consistent performance gains over established baselines, but does so with high statistical confidence across the majority of the STEM tasks.
>
> **W2. The authors train the gating mechanism/HRPO separately for each set of tasks.…**
>
> - We thank the reviewer’s concerns regarding the multi-task capabilities of HRPO. Our experiments follow previous works in reinforcement learning with verifiable rewards and latent reasoning, with similar training and evaluation datasets / settings [6-7]. Additionally, HRPO focuses on the algorithmic design of RL-based latent reasoning, and thus remains task‑agnostic and seamlessly supports multi‑task reinforcement learning. For example, we train a single model on the combined MMLU auxiliary dataset and ARC training set, then evaluate it on both benchmarks in our experiments. Similarly, we post-train one model solely on the NQ + HotpotQA data mixture and assess it on multiple knowledge-intensive tasks. Experiments show HRPO reaches the highest average EM with Qwen2.5-3B, outperforming the strongest RAG baseline by 4.3%. Similarly, HRPO with the smaller 1.5B backbone scores an average of 0.337, achieving consistent gains and over previous RL methods like PPO by 3.0%. These results demonstrate that HRPO can be easily extended to multi‑task scenarios similar to baseline RL algorithms like PPO or GRPO.
>
> **W3. The lack of a limitation section with a balanced discussion..…**
>
> - We acknowledge that further discussions on the limitations and impact statement could strengthen the paper. Therefore, we outline our key limitations and societal impact considerations which will be included in the revised manuscript. We hope these could address your concerns w.r.t. the limitations / impacts of HRPO.
>
> - **Limitations:** HRPO could introduce a few limitations compared to existing RL algorithms: it incurs additional computation and hyperparameter overhead due to the added gating parameters and the need to tune $r_{min}$ for optimal performance. In addition, its on‑policy design may reduce large-scale training efficiency compared to off‑policy methods. Moreover, although the hybrid outputs remain interpretable, the reasoning process happens in the continuous states and can be less clear due to unique patterns of HRPO (e.g., cross-lingual reasoning). In future works, we will explore simpler design, off‑policy extensions and advanced latent reasoning techniques to improve the interpretability, efficiency of HRPO and enhance RL-based continuous‑state reasoning.
>
> - **Societal Impacts:** HRPO itself is a task‑agnostic, algorithmic advance, similar to existing methods like PPO or GRPO, and does not by itself encode new domain‑specific behaviors. Its primary societal impact is therefore primarily the use of latent representations for reasoning. The positive impacts include HRPO’s adaptability, performance and concise reasoning, which could reduce compute costs and latency in complex AI systems. One potential risk is that, although latent reasoning remains interpretable, it could make reasoning traces less transparent and thus bring challenges in safety‑critical domains. Overall, we see no high‑level societal risks beyond those already posed by model‑based RL methods, but encourage continued discussions on the design / impacts of HRPO to ensure safety, alignment and transparency when deploying it for post‑training LLMs.
>
> **Q1. Why do GRPO and HRPO plots share almost all artifacts in Figure 3?**
>
> - In our experiments, we use the *identical random seeds and settings* (data, training steps, group / batch sizes etc.) for baseline methods and our HRPO. Given their shared components (reward function, advantage estimation etc.) between GRPO and HRPO, it is expected that GRPO and HRPO training reward curves exhibit very similar trends. As for the rapid early reward increase in HRPO, besides the initial warmup steps, we attribute this to HRPO’s gating component, which effectively learns hybrid representations and therefore yields faster convergence and higher reward plateaus.
>
> **Q2. Is there an experiment that could test the hypothesis in lines 268-270?**
>
> - The hypothesis that higher hidden ratios capture rich context information and shorten completions, can be validated by correlating hidden ratio trajectories with completion length and performance across different $r_{min}$ initializations, as we presented in Figure 4. This effect is more pronounced on knowledge‑intensive tasks, where we retrieve relevant contexts to answer complex multi-hop queries. For example, with the 1.5B backbone on HotpotQA, 2Wiki, and Bamboogle, HRPO’s average output lengths are 11.46, 99.99, and 93.09 tokens, respectively, versus GRPO’s 60.03, 101.82, and 101.93, suggesting consistent length reduction with increased latent context information.
>
> **Q3. What are the limitations of your submission?**
>
> - Please refer to our response to weakness 3.
>
> **Q4. What are the potential societal impacts of HRPO?**
>
> - Please refer to our response to weakness 3.
>
>
> [1] Sutton, Richard S., et al. "Policy gradient methods for reinforcement learning with function approximation." Advances in neural information processing systems 12 (1999).
>
> [2] Shao, Zhihong, et al. "Deepseekmath: Pushing the limits of mathematical reasoning in open language models." arXiv preprint arXiv:2402.03300 (2024).
>
> [3] Zeng, Weihao, et al. "Simplerl-zoo: Investigating and taming zero reinforcement learning for open base models in the wild." arXiv preprint arXiv:2503.18892 (2025).
>
> [4] Zuo, Yuxin, et al. "Ttrl: Test-time reinforcement learning." arXiv preprint arXiv:2504.16084 (2025).
>
> [5] Su, Yi, et al. "Crossing the Reward Bridge: Expanding RL with Verifiable Rewards Across
>
> [6] Hao, Shibo, et al. "Training large language models to reason in a continuous latent space." arXiv preprint arXiv:2412.06769 (2024).
>
> [7] Jin, Bowen, et al. "Search-r1: Training llms to reason and leverage search engines with reinforcement learning." arXiv preprint arXiv:2503.09516 (2025).

---

> > ### Comment · Reviewer_tnmr · 2025-08-06
> >
> > Thank you for your detailed and helpful rebuttal. You were able to resolve my main concerns and I will increase my score accordingly!

---

> ### Author Response · Authors · 2025-08-06
> **Discussion Reminder**
>
> We thank you for the time and effort you have dedicated to reviewing our submission! We hope our previous responses have addressed your concerns and remain available for any further discussions. Since the discussion period is ending, kindly let us know if you have further questions or concerns regarding our work. Thank you!

---

> ### Author Response · Authors · 2025-08-06
> **Response to Reviewer**
>
> We thank you again for the insightful feedback and your acknolwedgement on our rebuttal, we remain available in case you have any further questions regarding our submission, thanks!

---

> > ### Comment · Reviewer_tnmr · 2025-08-06
> >
> > Great! One thing I forgot to mention it explicitly, I increased the score to 5 and just assume that you would add the relevant clarifications to my questions, the limitation section, and the stat. significance tests in the final version of the paper.

---

> > > ### Author Response · Authors · 2025-08-06
> > > **Response to Reviewer**
> > >
> > > Thank you for your prompt response! We will ensure the additional experimental results and discussion are incorporated into our revised manuscript:)

---

### Official Review · Reviewer_AfNc · 2025-07-05

**Clarity:** 3
**Significance:** 3
**Originality:** 4
**Rating:** 5
**Confidence:** 4

**Summary:**

This paper introduces HRPO, a novel RL framework designed to enhance the reasoning abilities of LLM. The work primarily tackles key limitations of previous latent reasoning methods, like coconut, which demands extensive training resources and difficult to integrate with AR models. The core idea of this method is a learnable gating mechanism that dynamically blends the embeddings of discrete tokens with continuous hidden representations from previous steps. This enables the LLM to perform "hybrid reasoning". The model is trained with an on-policy RL framework using the simple rule-based reward. Extensive experiments on both knowledge-intensive and STEM datasets demonstrate the effectiveness of the method.

**Questions:**

1. `r_min` is firstly used on Line 267 without prior definitions.  Maybe I miss something. It is better to clarify the notation before using it.

2. It would be great if you can compare to coconut, despite that I know it is a bit of hard to well-reproduce their work.

3. More discussions (addition to Figure 4) about length reduction are appreciated. It is well-known that latent thinking is an efficient LLM reasoning method, which accelerates the reasoning process, both in faster speed and shorter length.

**Ethical Concerns:**

["NO or VERY MINOR ethics concerns only"]

**Final Justification:**

After carefully considering the authors' rebuttal and the discussions among reviewers, I maintain my original assessment of this submission.

**Limitations:**

yes

**Quality:**

3

**Strengths And Weaknesses:**

## Strengths
1. Good motivation and technically sound methodology. The central idea of using on-policy RL to train a model to learn a hybrid reasoning strategy is both novel and significant. It provides an elegant solution to the problem of integrating continuous reasoning into discrete LLMs without needing explicit CoT trajectories for supervision.

2. Comprehensive experiments and promising results.

## Weakness
1. Significance of improvements. While HRPO consistently outperforms GRPO, the margin of improvement is sometimes modest, particularly for the 3B model (e.g., 0.694 vs. 0.691 average on STEM, a 0.3% relative improvement; 0.378 vs. 0.367 average on Knowledge, a 1.1% improvement). While consistency across many datasets is a strong signal, a critic might argue the gains are incremental over a strong RL baseline.

2. Lack of theoretical analysis for the gating mechanism. For example, the Equation(4), I am confusing that why `a_t` is in such a formulation, why the weight between discrete token and latent thinking is balanced by `a_t` and `\sqrt{1-a_t^2}`, instead of `1 - a_t` or other formulations.

---

> ### Author Rebuttal · Authors · 2025-07-31
>
> We thank you for the constructive review and are excited that you find our work “novel”, “significant”, “good motivation” and “technically sound”. In the following, we have included responses and additional experimental results to respond to the raised weaknesses and questions, we hope these could address your concerns regarding our manuscript:
>
> **W1. Significance of improvements.…**
>
> - We acknowledge your concerns w.r.t. the statistical significance of the HRPO results in comparison to baseline methods. This stems from (i) our focus on a simple hybrid RL implementation using REINFORCE with group-relative rewards [1–2], which does not include optimization techniques like per-token rewards or ratio clipping; and (ii) we follow the standard practice of using greedy decoding for pass@1 evaluation [3–5], ensuring our results are easy to evaluate and reproducible. To address your concerns, we conduct additional sampling-based evaluations on the STEM tasks, which are often more challenging and exhibit greater variance. We present the averaged results below, with statistical significance (paired t-test) highlighted in bold:
>
> - | 1.5B | GSM8k | MATH  | MATH500 | MMLU-ST | ARC-C |
> | ------ | ----- | ----- | ------- | ------- | ----- |
> | PPO    | 0.701 | 0.505 | 0.511   | 0.551   | 0.716 |
> | GRPO   | 0.710 | 0.510 | 0.512   | 0.554   | 0.722 |
> | HRPO   | 0.712 | **0.515** | 0.517   | **0.565**   | **0.731** |
>
> - | 3B | GSM8k | MATH  | MATH500 | MMLU-ST | ARC-C |
> | ------ | ----- | ----- | ------- | ------- | ----- |
> | PPO    | 0.825 | 0.597 | 0.600   | 0.574   | 0.802 |
> | GRPO   | 0.827 | 0.595 | 0.599   | 0.577   | 0.808 |
> | HRPO   | **0.838** | **0.606** | **0.609**   | **0.585**   | 0.815 |
>
> - Across both backbones, HRPO outperforms PPO and GRPO on all benchmark datasets. For the 1.5B backbone HRPO, paired t-tests show these gains reach statistical significance (p < 0.05) in three out of five tasks. With the 3B model the improvements are even larger (+1.4%) respectively, and four out of five comparisons are statistically significant. These results demonstrate that our simple hybrid-RL framework HRPO not only delivers consistent performance gains over established baselines, but does so with high statistical confidence across the majority of the STEM tasks.
>
> **W2. Lack of theoretical analysis for the gating mechanism…**
>
> - We thank the reviewer’s interest in our gating mechanism. In the following, we explain why we mix the discrete token embeddings and the projected latent embeddings via the coefficients in Eq. 4. Our gating component is designed following common practice in state-space models / recurrent models [6-7], and the mixing is designed to have the following properties: (i) norm‑preserving circular mixing: hybrid embedding always lies on the same hypersphere as the original embeddings. This preserves the overall magnitude (and thus the scale/variance) of the input; (ii) spherical linear interpolation (slerp) between the two vectors. This yields a smooth arc on the unit sphere, rather than the straight‑line “lerp” that one would get with linear interpolation. Empirically, slerp‑style mixing maintains more consistent embedding geometry and leads to more stable RL rollouts, as demonstrated in our preliminary experiments and existing literature [8]. In brief, this choice ensures norm‑preservation and a smooth, non‑linear interpolation between the two representations. We will add a short theoretical rationale and the discussion of our design choice in the revised manuscript for clarification.
>
> **Q1. r_min is firstly used on Line 267 without prior definitions…**
>
> - We thank the reviewer for catching that $r_{min}$​ first appears in Line 267 without a prior definition. $r_{min}$ denotes the lower bound of the gating component’s sampling range; that is, during the initialization of $\Lambda$, we sample the gating coefficient $a_t$ uniformly from [$r_{min}$, 0.999]. With higher $r_{min}$​ values, this ensures that at initialization the model draws predominantly from token embeddings. To address this, we will introduce a clear definition of $r_{min}$ in our revised manuscript before the first occurrence of $r_{min}$.
>
> **Q2. It would be great if you can compare to coconut…**
>
> - Thank you for the suggestion, although HRPO represents the first hybrid latent reasoning framework via RL, we find it valuable to compare to other supervised latent reasoning methods such as Coconut (supervised CoT), CODI (supervised CoT + distillation) along with discrete-token CoT distilled from larger reasoning models [9-10]. We reported the full results in our technical appendix and provide these here for your reference. For GSM8k, Coconut achieves 0.315 accuracy, whereas HRPO reaches 0.720, a 40.5% improvement. Moreover, HRPO attains 0.518 accuracy on MATH, outperforming the strongest latent‑reasoning baseline by 10% (Coconut does not provide CoT split for MATH). These results show that, even though Coconut leverages hidden‑state feedback as “continuous thought,” it still lags significantly behind our RL‑based hybrid reasoning approach.
>
> - | Method         | GSM8k | MATH |
> | -------------- | ----- | ---- |
> | Coconut        | 0.315 | –    |
> | CODI           | 0.658 | 0.419|
> | Distilled CoT  | 0.706 | 0.503|
> | HRPO           | 0.720 | 0.518|
>
> **Q3. More discussions (addition to Figure 4) about length reduction…**
>
> We agree that further discussions on HRPO’s length‑reduction benefits could strengthen the paper. In the paper, we show that as training progresses, all variants’ completion lengths first grow and then contract, but HRPO with a smaller initial hidden‑ratio bound exhibits the steepest drop. This effect is more pronounced on knowledge‑intensive tasks, where we retrieve relevant contexts to answer complex multi-hop queries. For example, with the 1.5B backbone on HotpotQA, 2Wiki, and Bamboogle, HRPO’s average output lengths are 11.46, 99.99, and 93.09 tokens, respectively, versus GRPO’s 60.03, 101.82, and 101.93, suggesting consistent length reduction with increased latent context information. This may be traced back to the fact that the hidden representations can capture rich context information and thus shorten completions by providing a concise answer. We’ve also expanded this discussion in the revised manuscript and hope this can address your concerns.
>
>
> [1] Sutton, Richard S., et al. "Policy gradient methods for reinforcement learning with function approximation." Advances in neural information processing systems 12 (1999).
>
> [2] Shao, Zhihong, et al. "Deepseekmath: Pushing the limits of mathematical reasoning in open language models." arXiv preprint arXiv:2402.03300 (2024).
>
> [3] Zeng, Weihao, et al. "Simplerl-zoo: Investigating and taming zero reinforcement learning for open base models in the wild." arXiv preprint arXiv:2503.18892 (2025).
>
> [4] Zuo, Yuxin, et al. "Ttrl: Test-time reinforcement learning." arXiv preprint arXiv:2504.16084 (2025).
>
> [5] Su, Yi, et al. "Crossing the Reward Bridge: Expanding RL with Verifiable Rewards Across Diverse Domains." arXiv preprint arXiv:2503.23829 (2025).
>
> [6] Orvieto, Antonio, et al. "Resurrecting recurrent neural networks for long sequences." International Conference on Machine Learning. PMLR, 2023.
>
> [7] De, Soham, et al. "Griffin: Mixing gated linear recurrences with local attention for efficient language models." arXiv preprint arXiv:2402.19427 (2024).
>
> [8] Azuri, Idan, and Daphna Weinshall. "Generative latent implicit conditional optimization when learning from small sample." 2020 25th International Conference on Pattern Recognition (ICPR). IEEE, 2021.
>
> [9] Hao, Shibo, et al. "Training large language models to reason in a continuous latent space." arXiv preprint arXiv:2412.06769 (2024).
>
> [10] Shen, Zhenyi, et al. "Codi: Compressing chain-of-thought into continuous space via self-distillation." arXiv preprint arXiv:2502.21074 (2025).

---

> > ### Comment · Reviewer_AfNc · 2025-08-08
> >
> > Thank you for the thoughtful rebuttal. The authors answered all my question and addressed the concerns.

---

> > > ### Author Response · Authors · 2025-08-08
> > > **Response to Reviewer**
> > >
> > > Thank you for the thoughtful feedback and for recognizing that we have addressed all your questions and concerns. We appreciate your time and effort in reviewing our work!

---

### Note · Authors · 2025-08-13

We thank AC and all reviewers for their insightful feedback and constructive suggestions. We’re thrilled that reviewers find our work "novel", "elegant" and "technically sound"; regard our results as "promising", exhibit "intriguing behavior" and "superior performance". Below, we summarize our contributions, highlight the rebuttal efforts and reaffirm our commitments to improving the paper.

**Contributions**

- We introduce HRPO, the first RL framework for hybrid latent reasoning in LLMs. HRPO integrates discrete token embeddings and continuous hidden states through a gating mechanism, enabling latent-space RLVR without supervision. Our design introduces stochasticity into latent representations and overcomes the incompatibility with autoregressive generation, while preserving LLMs’ interpretability and efficiency. We validate HRPO via extensive experiments on knowledge and reasoning tasks, demonstrating that HRPO delivers consistent gains over strong RL baselines as well as supervised latent reasoning methods. These results highlight HRPO’s effectiveness, scalability and generalizability across domains.

**Rebuttal Efforts**

- We actively engaged in rebuttal to address raised concerns, including: (i) Statistical significance: provided t-test analysis showing that most gains are significant. (ii) Efficiency: elaborated on HRPO's efficiency and reported detailed metrics such as throughput, VRAM and latency. (iii) Methodology: expanded the rationale for HRPO’s design choices, such as its spherical interpolation in gating. (iv) Evaluation: incorporated additional results and analysis, such as comparisons with latent reasoning baselines. (v) Limitations: added discussions on constraints, potential risks and considerations for deploying HRPO in real-world scenarios.

- **Reviewer acknowledgement**: All reviewers confirmed that our rebuttal addressed their concerns. Reviewer K5az also indicated that they "accordingly adjusted the score". However, it appears the score was not updated in the system, and we note this solely to ensure the AC has full context in decision-making.

We hope these clarifications and additions highlight the novelty, technical quality and efficacy of our work. For revision, we will improve the paper’s presentation, expand methodology and experiment details in appendix, and include rebuttal discussions with supplementary results. We hope these could address the reviewers' concerns and improve the quality and clarity of our submission.

---

### Decision · Program_Chairs · 2025-09-17

**Decision:**

Accept (poster)

**Comment:**

The paper presents a novel RL-based approach for hybrid latent reasoning in LLMs. The method's key ingredient is a learnable gating mechanism that progressively integrates continuous hidden state representations from previous steps with discrete token embedding.

The proposed approach is novel and consistently improves upon alternatives on various benchmarks. The paper is well written and presents many valuable insights.

The authors have done an excellent job at addressing the reviewers concerns as appreciated by all reviewers. In particular they have provided statistical significance results, intuitive rationale and  details on the design of the gating mechanism, as well as computational efficiency result, and evaluation against Coconut.

We strongly urge the authors to incorporate all of the valuable materials and clarifying points into their manuscript.